# Phantom of Latent for Large Language and Vision Models

## Abstract

The success of visual instruction tuning has accelerated the development of large language and vision models (LLVMs). Following the scaling laws of instruction-tuned large language models (LLMs), LLVMs have also further increased in size, with examples including 26B, 34B, and even 80B parameters. While this increase in model size has yielded significant performance gains, it demands substantially more hardware resources for both training and inference. Consequently, there naturally exists a strong need for efficient LLVMs that achieve the performance of larger models while being smaller in size. To achieve this need, we present a new efficient LLVM family with model sizes of 0.5B, 1.8B, 3.8B, and 7B parameters, 👻 **Phantom**, which significantly enhances learning capabilities within limited structures. By temporarily increasing the latent hidden dimension during multi-head self-attention (MHSA), we make LLVMs understand much more vision-language knowledge on the latent, without substantially increasing physical model sizes. To maximize its advantage, we introduce **P**hantom **O**ptimization (**PO**) using both autoregressive supervised fine-tuning (SFT) and direct preference optimization (DPO)-like concept, which effectively follows correct answers while eliminating incorrect and ambiguous ones. 👻 **Phantom** outperforms numerous larger open- and closed-source LLVMs, positioning itself as a leading solution in the landscape of efficient LLVMs.

## 1 Introduction

In recent years, artificial general intelligence (AGI) has increasingly become a part of daily life, significantly enhancing our convenience. This trend is largely attributed to technical advancements of large language models (LLMs) and their impressive generalization performance, facilitated by instruction tuning (Wei et al., 2022; Chung et al., 2022). Building on this momentum, instruction tuning has expanded its realm into visual instruction tuning (Liu et al., 2023c), integrating both language and vision as a format of text and image, under the use of pretrained LLMs. Based on them, numerous large language and vision models (LLVMs) have continuously emerged as multimodal LLMs and they have shown outstanding vision-language performances.

In terms of open-to-public regarding model architectures and their trained parameters, LLVMs can be categorized into open-source and closed-source models. For example, there are representative closed ones: GPT-4V (OpenAI, 2023), Gemini-Pro (Team et al., 2023), and Qwen-VL-Plus (Bai et al., 2023a;b), all of which are renowned for their remarkable vision-language performances, large model sizes, and extensive number of dataset samples. In response, open-source LLVMs have tried to narrow the performance gap with their closed-source performances, by following the similar strategies the closed ones used, such as scaling up model sizes (Liu et al., 2024a; McKinzie et al., 2024; Li et al., 2024d) (*e.g.,* 26B, 34B, and 80B) and curating larger number of visual instruction tuning samples (Hu et al., 2024a; Fang et al., 2024; Tong et al., 2024) (*e.g.,* 4M, 6M, and 10M).

Along with them, several modules have focused on image-level understanding by leveraging numerous types of vision encoders (Kar et al., 2024; Lu et al., 2024; Goncharova et al., 2024; Ranzinger et al., 2023; Zhao et al., 2024; Li et al., 2024d) and multiple computer vision models (Chen et al., 2024a; Wang et al., 2024c; Jiao et al., 2024; Lee et al., 2024c;d). Additionally, a series of projectors have been employed alongside various vision encoders to improve fine-grained understanding (Li

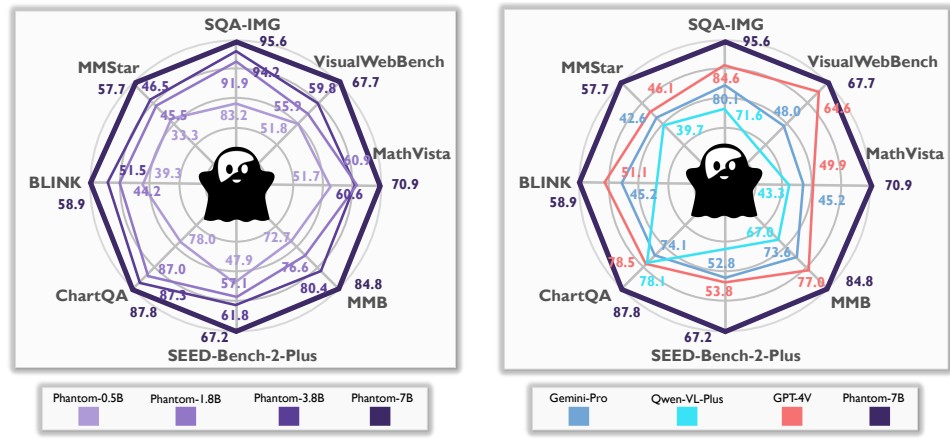

Figure 1: Overview of performances compared with 👻 Phantom and closed-source LLVMs

et al., 2024d; Tong et al., 2024; Ge et al., 2024a; Chen et al., 2024c; Yao et al., 2024) through partitioning the image. Besides, a multifaceted rationale-embedded projector (Lee et al., 2024b) has been used to enhance real-world knowledge such as document, chart, and math.

However, these efforts — summarized as (a) scaling up model size, (b) curating larger datasets, and (c) incorporating additional modules and projectors — may not be regarded as a primary key to basically improve their own learning capabilities of LLVMs. In other words, there remains unexplored potential in fully utilizing LLVMs to align vision knowledge with language one and embed much more vision-language knowledge within limited structures, without relying on external modules and projectors. Beyond their limited learning capabilities, specifically, (a) and (b) bring in striking computational burdens during training, necessitating high-end GPUs with substantial VRAM. This (a) more becomes a critical drawback in devices with limited GPU resources, such as mobile phones and embedded boards. Furthermore, the high computational inference costs, associated with larger model sizes, exacerbate these issues, particularly for real-time applications such as augmented reality (AR) systems. As a result, deploying and operating LLVMs in such resource-constrained on-device environments becomes a major challenge.

To meet the two needs of maintaining model sizes while achieving superior performance, we present an efficient LLVM family, 👻 **Phantom**, which stimulates enlarging vision-language learning capabilities within limited structures. When conducting multi-head self-attention (MHSA), 👻 Phantom temporarily increases the latent hidden dimension and understand much more vision-language knowledge. Without significantly increasing the physical model size, we get an effect of increasing the dimension in query, key, and value, which we now call as Phantom Dimension. In order to maximally boost this advantage, we introduce *Phantom Optimization (PO)*, inspired by RLHF and DPO (Christiano et al., 2017; Stiennon et al., 2020; Ouyang et al., 2022; Rafailov et al., 2024; Hong et al., 2024a; Meng et al., 2024). Unlike traditional preference-based methods, PO is designed to minimize the generation of incorrect and ambiguous answers. Since autoregressive supervised fine-tuning (SFT) primarily focuses on producing correct answers, PO provides 👻 Phantom with additional guidance to avoid confusing answers by borrowing the recent DPO formulation (Meng et al., 2024).

To do so, we first need a collection of incorrect and ambiguous answers. These are generated and filtered through GPT-4o(-mini) and human review from 2.8M visual instruction tuning samples covering diverse capabilities (details in Section 3). This process resulted in the curation of 2M Phantom triples including question, its correct answer, and the corresponding incorrect and ambiguous answers (see Appendix A). By using the triple, 👻 Phantom is trained with the two training steps, where we train vision projector and Phantom Dimension in the first step with the pretrained LLM frozen. In the second step, all components are trained together. Notably, PO utilizes SFT together with DPO-like concept throughout first training step, making 👻 Phantom have an ability that follows correct answers while eliminating incorrect and ambiguous ones. In the experiment section, we demonstrate that handling the latent hidden dimension and using PO enhances vision-language performances by a large margin. As a result, we release an efficient LLVM family 👻 Phantom with 0.5B, 1.8B, 3.8B, and 7B model sizes, which outperform open- and closed-source LLVMs, establishing a leading solution in the realm of efficient LLVMs.

Our contribution can be summarized into two main aspects:

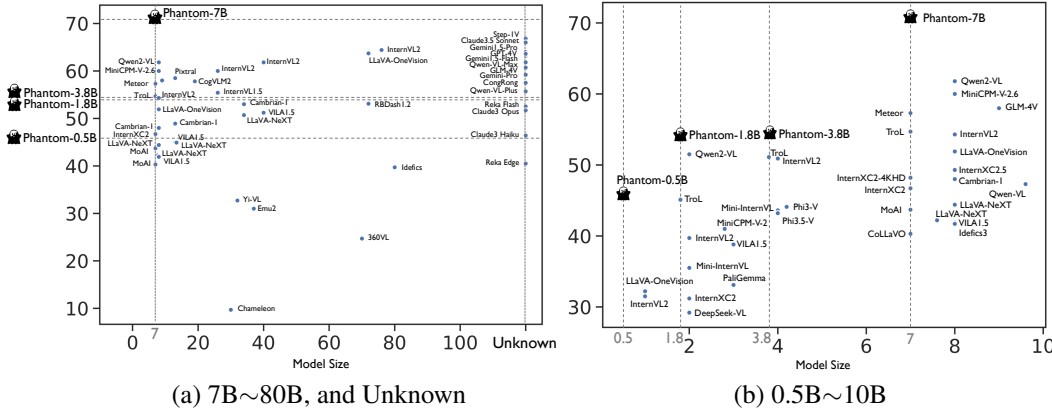

(a) 7B∼80B, and Unknown         (b) 0.5B∼10B

Figure 2: Evaluating MM-Vet (Yu et al., 2023) for efficient LLVM family, ♟ Phantom, across four model sizes (0.5B, 1.8B, 3.8B, and 7B), compared with various model size LLVMs: (a) 7B∼80B and unknown model size for closed-source LLVMs (b) 0.5B∼10B model sizes.

- We present a new efficient large language and vision model (LLVM) Family, ♟ **Phantom**, which temporarily increases the latent hidden dimension during multi-head self-attention (MHSA) to enhance vision-language learning capabilities within limited structures.

- Curating efficient size 2M number of Phantom triples, we introduce a training strategy of **P**hantom **O**ptimization (**PO**) which avoids incorrect and ambiguous answers, showcasing more advancements across numerous evaluation benchmarks.

## 2 RELATED WORKS

**Large Language and Vision Models.** To bridge the performance gap with closed-source LLVMs, open-source LLVMs have adopted three primary strategies: scaling up model size, curating larger datasets, and incorporating additional modules or projectors. For instance, LLaVA-NeXT (Liu et al., 2024a), MM1 (McKinzie et al., 2024), Yi-VL (Young et al., 2024) and MiniGemini (Li et al., 2024d) build model variants with parameters up to 34B. Concurrent to these efforts, mPLUG-Owl (Hu et al., 2024a), VILA[2] (Fang et al., 2024), and Cambrian-1 (Tong et al., 2024) curate high-quality visual instruction tuning datasets specialized for diverse visual capabilities. Lastly, recent works have leveraged various vision encoders (Kar et al., 2024; Lu et al., 2024; Goncharova et al., 2024; Ranzinger et al., 2023; Zhao et al., 2024; Li et al., 2024d) and integrated external computer vision modules (Chen et al., 2024a; Wang et al., 2024c; Jiao et al., 2024; Lee et al., 2024c;d) to expand LLVMs' perception capabilities. Alongside using extra vision encoders, several works utilize projectors to extract hierarchical features of images (Li et al., 2024d; Tong et al., 2024; Ge et al., 2024a; Chen et al., 2024c; Yao et al., 2024) or to improve real-world knowledge comprehension such as document analysis, chart interpretation, and mathematical reasoning (Lee et al., 2024b).

While these approaches enhance downstream task performance, they do not address the core challenge of improving the intrinsic learning capabilities of LLVMs. Scaling up model size or employing larger instruction tuning datasets leads to substantial computational burdens. In addition, relying on extra visual encoders or computer vision modules brings in external visual knowledge, but they mainly focus on visual perception-related capabilities and their additional parameters may also lead the burden. This underscores the need for developing more efficient LLVMs with enhanced inherent capabilities that do not depend on such resource-intensive strategies.

**Efficient Modeling.** In an effort to enhance the fundamental capabilities of LLMs while maintaining model size, several works for natural language processing has increasingly focused on developing smaller model sizes (Thawakar et al., 2024; Mehta et al., 2024; Liu et al., 2024c), network pruning (Ma et al., 2023; Men et al., 2024; Ashkboos et al., 2024), and quantization (Li et al., 2023c; Shao et al., 2024a; Park et al., 2024a). These approaches primarily aim to accelerate training speed and reduce inference time while retaining performance, rather than boosting performances or improving LLVMs' embedding capabilities of vision-language knowledge within the limited structures. While efficient modeling has been extensively explored for LLMs, the design of efficient vision-language models (LLVMs) remains underexplored. A recent work, TroL (Lee et al., 2024a), uniquely introduces a layer traversing technique that reuses layers in a token-wise manner to potentially embed more vision-language knowledge. However, it faces significant challenges, such as

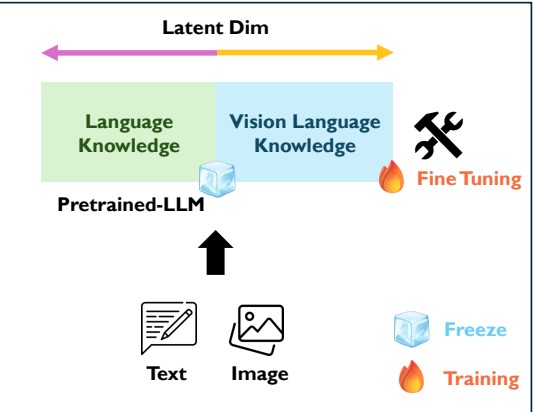

(a) Previous Training for LLVMs       (b) Training LLVMs with Phantom Dimension

Figure 3: Comparing training paradigm for building (a) previous LLVMs and (b) 👻 Phantom equipped with Phantom Dimension

increased inference time due to doubling layer propagation and critical issues with key-value cache storage, preventing it from fully realizing its potential for efficient LLVMs.

In response to the need for efficient yet high-performing LLVMs, we introduce a new efficient LLVM family, 👻 **Phantom**, which enhances the embedding capability of vision-language knowledge by temporarily increasing the latent hidden dimension during multi-head self-attention (MHSA). This innovation, combined with 2M Phantom triples to guide LLVMs towards correct answers while avoiding confusion, is expected to pave the way for more efficient LLVMs in both training and inference and to represent a crucial first step in advancing the field.

## 3    👻   PHANTOM

**Existing LLVMs and 👻 Phantom.** Figure 3(a) depicts the commonly used training paradigm for building LLVMs, where pretrained LLMs are fine-tuned to acquire visual understanding and handle vision-language tasks using both text and image inputs. This approach directly modifies the original latent dimensions derived from the their parameters to accommodate the new vision-language capabilities. In contrast, Figure 3(b) introduces a new concept of expanding the latent dimension, which we call as Phantom Dimension, which trains only the added latent space without fine-tuning the entire pretrained LLMs. It enables for pretrained LLMs to have a room to embed the new knowledge, which effectively integrates vision language knowledge without overwritting the parameters and altering or compromising the original knowledge. In other words, Phantom Dimension injects vision language knowledge to the pretrained-LLMs directly with the enlarged hidden dimension, where it can be considered as another hidden room of internal model architectures. By leveraging this approach, we build a new LLVM family of 👻 Phantom and in order to extremely boost its vision language performances, we curate incorrect and ambiguous answers and present a way utilizing them, called as Phantom Optimization. Next, we will explain model architecture of 👻 Phantom, and the realization of Phantom Dimension. In addition, the process of creating incorrect and ambiguous answers, as well as strategies for effectively utilizing them, will be discussed.

**Overview of Model Architecture.** As shown in Figure 4(a), the architecture of 👻 Phantom model consists of vision encoder, vision projector, and a multimodal language model including word embedding and language model head, which follows a common configuration used in open-source LLVMs (Liu et al., 2023c;b; Bai et al., 2023b; Chen et al., 2023a; McKinzie et al., 2024). Specifically, we utilize InternViT-300M (Chen et al., 2023b) as the vision encoder instead of CLIP-L-428M (Radford et al., 2021), due to its superior ability to align text-to-image representations through contrastive learning with large language models (LLMs). The vision projector is constructed using two fully connected layers, where GELU (Hendrycks & Gimpel, 2016) activation function is interleaved with each layer. For multimodal LLM component, we initialize it using pretrained LLMs across various sizes, selected for their state-of-the-art performance within their respective size: Qwen2-0.5B (Yang et al., 2024), InternLM2-1.8B (Cai et al., 2024), Phi3-mini-3.8B Abdin et al. (2024), and InternLM2.5-7B (Cai et al., 2024).

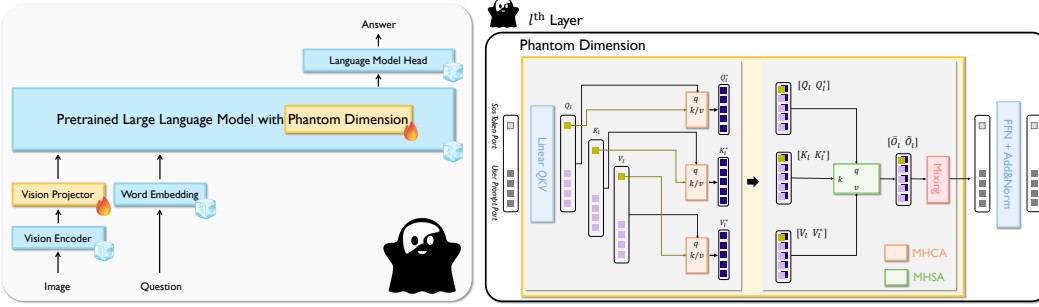

(a) Model Architecture        (b) Phantom Dimension

Figure 4: (a) Overview of model architecture and the detail of first training step with Phantom Dimension and Phantom Optimization. In second training step, we train all of the parameters described in this figure. (b) Illuminating how Phantom Dimension temporarily enlarges the latent hidden dimension in forward propagation at $l$-th layer in 👻 Phantom, where 'Linear QKV', MHSA, and 'FFN+Add&Norm' is generally used module from pretrained LLM. Only MHCA module is added.

---

**Algorithm 1** A Transformer Decoder Block with Phantom Dimension

---

1: **Input:** $X = [x_{\text{sos}}, x_{\text{prompt}}]$
2: $Q_l, K_l, V_l \leftarrow \textbf{LinearQKV}(X)$
3: $Q_l^*, K_l^*, V_l^* \leftarrow Q_l[0], K_l[0], V_l[0]$ {Extracting feature for the place of 'sos' token}
4: $Q_l^*, K_l^*, V_l^* \leftarrow \text{MHCA}(Q_l, Q_l^*), \text{MHCA}(K_l, K_l^*), \text{MHCA}(V_l, V_l^*)$
5: $[\bar{O}_l, \tilde{O}_l] \leftarrow \textbf{MHSA}(q = [Q_l, Q_l^*], k = [K_l, K_l^*], v = [V_l, V_l^*])$
6: $\bar{w}, \tilde{w} \leftarrow e^{f(\bar{O}_l)}/(e^{f(\bar{O}_l)} + e^{g(\tilde{O}_l)}), e^{g(\tilde{O}_l)}/(e^{f(\bar{O}_l)} + e^{g(\tilde{O}_l)})$
7: $O_l \leftarrow \bar{w} \odot \bar{O}_l + \tilde{w} \odot \tilde{O}_l$
8: $X \leftarrow \textbf{FFN+Add\&Norm}(O_l)$
9: **Return:** $X$

---

**Realization of Phantom Dimension.** For better understanding, Figure 4(b) represents the simple overview of how Phantom Dimension works. We utilize start of sequence (sos) token that will serve as a key in enhancing the latent hidden dimension for the query, key, and value components in multi-head self-attention (MHSA) layers. The latent feature on the location of sos token is propagated into QKV linear function, and we denote its outputs as $Q_l^* \in \mathbb{R}^{d_q}$, $K_l^* \in \mathbb{R}^{d_{kv}}$, and $V_l^* \in \mathbb{R}^{d_{kv}}$ at each layer $l$. Note that $d$ denotes the latent hidden dimension. $Q_l^*$, $K_l^*$, and $V_l^* \in \mathbb{R}^{d_{kv}}$ are supposed to inject into the multi-head cross-attention (MHCA) module. A natural question arises: *Why inject these features into the cross-attention module?* The reason lies in the dynamic length $N$ of user input tokens, which varies with the question length. Therefore, these features need to have dimension $Q_l^* \in \mathbb{R}^{N \times d_q}$, $K_l^* \in \mathbb{R}^{N \times d_{kv}}$, and $V_l^* \in \mathbb{R}^{N \times d_{kv}}$ since sos token only represents a single token. Therefore, it must be expanded to match the $N$ tokens of the input sequence, and the cross-attention module make these features expanded into input sequence token number $N$, as follows:

$$
\begin{aligned}
Q_l^* &\leftarrow \text{MHCA}(q = Q_l, k/v = Q_l^*), \\
K_l^* &\leftarrow \text{MHCA}(q = K_l, k/v = K_l^*), \\
V_l^* &\leftarrow \text{MHCA}(q = V_l, k/v = V_l^*),
\end{aligned}
\tag{1}
$$

where we change their dimension into $Q_l$: $\mathbb{R}^{N \times h_q \times \frac{d_q}{h_q}}$ and $K_l, V_l$: $\mathbb{R}^{h_{kv} \times \frac{d_{kv}}{h_{kv}}}$ for conducting multi-head cross attention with head number $h_q$ and $h_{kv}$. Next, in order to make LLVMs embed much more vision-language knowledge, we enlarge the latent hidden dimension by concatenating the original query, key, and value matrices with the cross-attended outputs dimension-wise, yielding $[Q_l \ \ Q_l^*] \in \mathbb{R}^{N \times h_q \times \frac{2d_q}{h_q}}$, $[K_l \ \ K_l^*] \in \mathbb{R}^{N \times h_{kv} \times \frac{2d_{kv}}{h_{kv}}}$, and $[V_l \ \ V_l^*] \in \mathbb{R}^{N \times h_{kv} \times \frac{2d_{kv}}{h_{kv}}}$. We then apply multi-head self-attention (MHSA) used in multimodal LLM to these concatenated ones:

$$
O_l = \text{Softmax}\left( \lambda \left( \frac{2d_q}{h_q} \right)^{-\frac{1}{2}} [Q_l \ \ Q_l^*][K_l \ \ K_l^*]^\top \right) [V_l \ \ V_l^*],
\tag{2}
$$

where $\lambda$ denotes a regularization parameter, and $O_l \in \mathbb{R}^{N \times h_q \times \frac{2d_q}{h_q}}$ represents the output features of MHSA. After its computation, the output features should return to the original hidden dimension, as they will be propagated through the remaining transformer modules, such as feed-forward network

Table 1: Comparison with the current existing standard model size open-source LLVMs, evaluating vision-language performances of 📷 Phantom on numerous general evaluation benchmarks: SQA[I] (Lu et al., 2022), AI2D (Kembhavi et al., 2016), ChartQA (Masry et al., 2022), SEED[I] (Li et al., 2023a), POPE (Li et al., 2023b), HallB (Liu et al., 2023a), MME (Fu et al., 2023), MathVista (Lu et al., 2023), MMB (Liu et al., 2023d), MMB[CN] (Liu et al., 2023d), MM-Vet (Yu et al., 2023), and LLaVA[W] (Liu et al., 2023c). **Bold** and Underline represent the top and the second, each.

| LLVMs | SQA[I] | AI2D | ChartQA | SEED[I] | POPE | HallB | MME | MathVista | MMB | MMB[CN] | MM-Vet | LLaVA[W] |
|---|---|---|---|---|---|---|---|---|---|---|---|---|
| ShareGPT4V-7B (Chen et al., 2023a) | 68.4 | - | - | 69.7 | - | 49.8 | 1944 | 25.8 | 68.8 | 62.2 | 37.6 | - |
| InternLM-XC-7B (Zhang et al., 2023) | - | - | - | 66.1 | - | 57.0 | 1919 | 29.5 | 74.4 | 72.4 | 35.2 | - |
| Monkey-10B (Li et al., 2023d) | 69.4 | - | - | 68.9 | - | 58.4 | 1924 | 34.8 | 72.4 | 67.5 | 33.0 | - |
| VILA-7B (Lin et al., 2023a) | 68.2 | - | - | 61.1 | 85.5 | - | - | - | 68.9 | 61.7 | 34.9 | - |
| VILA-13B (Lin et al., 2023a) | 73.7 | - | - | 62.8 | 84.2 | - | - | - | 70.3 | 64.3 | 38.8 | - |
| SPHINX-7B (Lin et al., 2023b) | 70.6 | - | - | 71.6 | 86.9 | - | 1797 | 27.8 | 65.9 | 57.9 | 40.2 | - |
| SPHINX-MoE-7B×8 (Gao et al., 2024) | 70.6 | - | - | 73.0 | **89.6** | - | 1852 | 42.7 | 71.3 | - | 40.9 | - |
| SPHINX-Plus-13B (Gao et al., 2024) | 70.6 | - | - | 74.8 | 89.1 | 52.1 | 1741 | 36.8 | 71.0 | - | 47.9 | - |
| LLaVA-NeXT-7B (Liu et al., 2024a) | 70.1 | - | - | 70.2 | 86.5 | - | 1851 | 34.6 | 69.6 | 63.3 | 43.9 | 72.3 |
| LLaVA-NeXT-8B (Liu et al., 2024a) | - | 71.6 | 69.5 | - | - | - | 1972 | 37.5 | 72.1 | - | - | 80.1 |
| LLaVA-NeXT-13B (Liu et al., 2024a) | 73.6 | 70.0 | 62.2 | 72.2 | 86.7 | - | 1892 | 35.1 | 70.0 | 68.5 | 47.3 | 72.3 |
| MM1-7B (McKinzie et al., 2024) | 72.6 | - | - | 69.9 | 86.6 | - | 1858 | 35.9 | 72.3 | - | 42.1 | - |
| MM1-MoE-7B×32 (McKinzie et al., 2024) | 74.4 | - | - | 70.9 | 87.8 | - | 1992 | 40.9 | 72.7 | - | 45.2 | - |
| MiniGemini-HD-7B (Li et al., 2024d) | - | - | - | - | - | - | 1865 | 32.2 | 65.8 | - | 41.3 | - |
| MiniGemini-HD-13B (Li et al., 2024d) | - | - | - | - | - | - | 1917 | 37.0 | 68.6 | - | 50.5 | - |
| Cambrian-1-8B (Tong et al., 2024) | 80.4 | 73.0 | 73.3 | 74.7 | - | - | - | 49.0 | 75.9 | - | - | - |
| Cambrian-1-13B (Tong et al., 2024) | 79.3 | 73.6 | 73.8 | 74.4 | - | - | - | 48.0 | 75.7 | - | - | - |
| Eagle-8B (Shi et al., 2024) | 84.3 | 76.1 | 80.1 | 76.3 | - | - | - | 52.7 | 75.9 | - | - | - |
| Eagle-13B (Shi et al., 2024) | 82.0 | 74.0 | 77.6 | 74.8 | - | - | - | 54.4 | 75.7 | - | - | - |
| VILA1.5-8B (Lin et al., 2023a) | 82.0 | - | - | 73.8 | 85.6 | - | - | - | 75.3 | 69.9 | 43.2 | 71.9 |
| VILA1.5-13B (Lin et al., 2023a) | 80.1 | - | - | 72.6 | 86.3 | - | - | - | 74.9 | 66.3 | 44.3 | 80.8 |
| VILA[2]-8B (Fang et al., 2024) | 87.6 | - | - | 66.1 | 86.7 | - | - | - | 76.6 | 71.7 | 50.0 | 86.6 |
| MiniCPM-V-2.5-8B (Yao et al., 2024) | - | - | - | - | - | - | 2025 | 54.3 | 77.2 | 74.2 | - | 86.7 |
| CogVLM2-8B (Hong et al., 2024b) | - | 73.4 | 81.0 | - | - | - | 1870 | - | 80.5 | - | 60.4 | - |
| TroL-7B (Lee et al., 2024a) | **92.8** | 78.5 | 71.2 | 75.3 | 87.8 | 65.3 | **2308** | - | 83.5 | 81.2 | 54.7 | **92.8** |
| Molmo-7B (Deitke et al., 2024) | 92.2 | 79.6 | 84.1 | 74.1 | 87.4 | - | 1784 | 47.3 | - | - | 53.3 | 74.0 |
| LLaVA-OneVision-8B (Li et al., 2024a) | **96.0** | **81.4** | 80.0 | **75.4** | - | - | 1998 | 63.2 | 80.8 | - | 57.5 | 90.7 |
| 📷 Phantom-7B | 95.6 | 79.5 | **87.8** | 75.3 | 87.7 | **65.4** | 2126 | 70.9 | **84.8** | **84.7** | **70.8** | 84.9 |

(FFN). At this stage, we aim to compress the output features while minimizing information loss as much as possible. To achieve this, we split the output $O_l$ into two halves: $O_l[:, :, :\frac{d_q}{h_q}]$ and $O_l[:, :, \frac{d_q}{h_q}:]$ (Python slicing format), denoted as $\bar{O}_l \in \mathbb{R}^{N \times h_q \times \frac{d_q}{h_q}}$ and $\tilde{O}_l \in \mathbb{R}^{N \times h_q \times \frac{d_q}{h_q}}$, respectively. To flexibly mix them, weighted-average operation is employed, and then finally we can get the compressed outputs $O_l \leftarrow \bar{w} \odot \bar{O}_l + \tilde{w} \odot \tilde{O}_l$ where $\odot$ is element-wise multiplication, and

$$\bar{w} = \frac{e^{f(\bar{O}_l)}}{e^{f(\bar{O}_l)} + e^{g(\tilde{O}_l)}}, \quad \tilde{w} = \frac{e^{g(\tilde{O}_l)}}{e^{f(\bar{O}_l)} + e^{g(\tilde{O}_l)}}, \tag{3}$$

where $f$ and $g$ comprise each one fully-connected layer: $\mathbb{R}^{N \times h_q \times \frac{d_q}{h_q}} \to \mathbb{R}^{N \times h_q}$, and the compressed outputs are then propagated into remaining modules with root mean square (RMS) layer normalization (Ba et al., 2016; Zhang & Sennrich, 2019) and Add&Norm operation. Algorithm 1 shows a brief summary for forward propagation procedure of a transformer decoder block equipped with Phantom Dimension, where **Bold** means the original modules normally used in transformer decoder block and Orange represents the added modules from Phantom Dimension. All of the modules independently exist across every layers.

**Curation of Phantom Triples.** From 2.8M visual instruction tuning samples (see Appendix B), we generate incorrect and ambiguous answers based on the existing question-answer pairs. To reduce data generation costs, we utilize GPT-4o-mini with the following prompt: *"Question:* {}. *Answer:* {}. *Based on the question and the answer, make an incorrect and ambiguous answer compared to the original one. The length of the original answer should be maintained. Do not include any additional text."*. Here, {} serves as a placeholder. Next, we employ GPT-4o to validate the generated responses using the prompt: *"Original Answer :* {}. *Incorrect and Ambiguous Answer:* {}. *Provide 'Yes' or 'No', where 'Yes' means it is incorrect and ambiguous answer compared to the original one, 'No' means it is correct answer compared to the original one. Do not include any additional text."*. All samples labeled 'No' are discarded, while the 'Yes'-labeled samples undergo human review to verify if they are genuinely confusing. Through this process, we curate 2M Phantom Triples, consisting of a question, its correct answer, and a corresponding confusing answer.

**Implementation of Phantom Optimization.** To fully leverage the enhanced learning capability provided by Phantom Dimension, we introduce Phantom Optimization (PO), which is heavily inspired by Direct Preference Optimization (DPO) (Rafailov et al., 2024). While methods such as

Table 2: Comparison with the current existing smaller open-source LLVMs across 0.5B∼4B model sizes, evaluating vision-language performances of 🎬 Phantom on numerous evaluation benchmarks equally used in Table 1.

| LLVMs | SQA[I] | AI2D | ChartQA | SEED[I] | POPE | HallB | MME | MathVista | MMB | MMB[CN] | MM-Vet | LLaVA[W] |
|---|---|---|---|---|---|---|---|---|---|---|---|---|
| MobileVLM-3B (Chu et al., 2023) | 61.2 | - | - | - | 84.9 | - | - | - | 59.6 | - | - | - |
| MobileVLM-V2-3B (Chu et al., 2024) | 70.0 | - | - | - | 84.7 | - | - | - | 63.2 | - | - | - |
| MoE-LLaVA-2.7B×4 (Lin et al., 2024) | 70.3 | - | - | - | 85.7 | - | - | - | 68.0 | - | 35.9 | - |
| LLaVA-Phi-2.7B (Zhu et al., 2024) | 68.4 | - | - | - | 85.0 | - | - | - | 59.8 | - | 28.9 | - |
| Imp-v1-3B (Shao et al., 2024b) | 70.0 | - | - | - | **88.0** | - | - | - | 66.5 | - | 33.1 | - |
| TinyLLaVA-3.1B (Zhou et al., 2024) | 69.1 | - | - | - | 86.4 | - | - | - | 66.9 | - | 32.0 | - |
| TinyLLaVA-Sig-Phi-3.1B (Zhou et al., 2024) | 69.1 | - | - | - | 86.4 | - | - | - | 66.9 | - | 32.0 | - |
| Bunny-3B (He et al., 2024) | 70.9 | 38.2 | - | 62.5 | 86.8 | - | 1778 | - | 68.6 | - | - | - |
| MiniCPM-2.4B (Hu et al., 2024b) | - | 56.3 | - | - | - | - | 1650 | 28.9 | 64.1 | 62.6 | 31.1 | - |
| MiniCPM-V2-2.8B (Hu et al., 2024b) | - | 62.9 | - | - | - | - | 1809 | 38.7 | 69.1 | 66.5 | 41.0 | - |
| MM1-3B (McKinzie et al., 2024) | 69.4 | - | - | 68.8 | 87.4 | - | 1762 | 32.0 | 67.8 | - | 43.7 | - |
| MM1-MoE-3B×64 (McKinzie et al., 2024) | 76.1 | - | - | 69.4 | 87.6 | - | 1773 | 32.6 | 70.8 | - | 42.2 | - |
| ALLaVA-3B (Chen et al., 2024b) | - | - | - | 65.2 | - | - | 1623 | - | 64.0 | - | 32.2 | - |
| ALLaVA-3B-Longer (Chen et al., 2024b) | - | - | - | 65.6 | - | - | 1564 | - | 64.6 | - | 35.5 | - |
| VILA1.5-3B (Chen et al., 2024b) | 69.6 | - | - | 66.4 | 85.3 | - | - | - | 62.8 | 52.2 | 38.6 | **76.7** |
| TroL-3.8B (Lee et al., 2024a) | 90.8 | 73.6 | 73.8 | 70.5 | 86.5 | 62.2 | 1980 | 55.1 | 79.2 | 77.1 | 51.1 | 76.6 |
| 🎬 Phantom-3.8B | **94.2** | 71.7 | 87.3 | 72.8 | 87.1 | 60.8 | 2046 | 60.6 | 80.4 | 77.1 | 54.4 | 76.2 |
| DeepSeek-VL-1.3B (Lu et al., 2024) | - | - | - | 66.7 | 87.6 | - | - | 31.1 | 64.6 | 62.9 | 34.8 | - |
| MobileVLM-1.7B (Chu et al., 2023) | 57.3 | - | - | - | 84.5 | - | - | - | 53.2 | - | - | - |
| MobileVLM-V2-1.7B (Chu et al., 2024) | 66.7 | - | - | - | 84.3 | - | - | - | 57.7 | - | - | - |
| MoE-LLaVA-1.8B×4 (Lin et al., 2024) | 63.1 | - | - | - | 87.0 | - | - | - | 59.7 | - | 25.3 | - |
| Mini-Gemini-2B (Li et al., 2024d) | - | - | - | - | - | - | 1653 | 29.4 | 59.8 | - | - | - |
| TroL-1.8B (Lee et al., 2024a) | 87.5 | 68.9 | 64.0 | 69.0 | 88.6 | 60.1 | 2038 | 45.4 | 76.1 | 74.1 | 45.1 | **69.7** |
| 🎬 Phantom-1.8B | 91.9 | 62.3 | 87.0 | 68.6 | 89.6 | 62.2 | 1885 | 60.9 | 76.6 | 75.1 | 54.1 | 68.6 |
| LLaVA-OneVision-0.5B (Li et al., 2024a) | 67.2 | **57.1** | 61.4 | **65.5** | - | - | 1478 | 34.8 | 52.1 | - | 29.1 | **74.2** |
| 🎬 Phantom-0.5B | **83.2** | 54.1 | 78.0 | 60.6 | 86.0 | 54.6 | 1743 | 51.7 | 72.7 | 70.1 | 45.7 | 69.6 |

RLHF (Christiano et al., 2017) and DPO are designed to optimize towards human or AI-driven preferences, PO is tailored to follow correct answer and reduce incorrect and ambiguous answers during training. To reduce the computational complexity of incorporating an additional reference model, we adopt the loss formulation from SimPO (Meng et al., 2024). Similar to ORPO (Hong et al., 2024a), we simultaneously use autoregressive supervised fine-tuning (SFT). This enables 🎬 Phantom to effectively reinforce correct answers $y^+$ while eliminating incorrect and ambiguous ones $y^-$ in response to a given prompt $x$. This formulation can be expressed as follows:

$$\min_\theta \mathcal{L}_{\text{PO}} = \mathcal{L}_{\text{SFT}} - \mathbb{E}_\mathcal{D}\left[\log\sigma\left(\frac{\beta}{|y^+|}\log\pi_\theta(y^+|x) - \frac{\beta}{|y^-|}\log\pi_\theta(y^-|x) - \gamma\right)\right], \qquad (4)$$

where $\theta$ represents the trainable parameters and $\mathcal{L}_{\text{SFT}}$ denotes the supervised fine-tuning loss for question-answer pairs. Note that, $\mathcal{D}$ is a dataset for Phantom triples $(x, y^+, y^-) \sim \mathcal{D}$, $\sigma$ denotes sigmoid function, $\beta$ and $\gamma$ means the hyper-parameter used in SimPO (Meng et al., 2024), where if $\beta$ increases, then the gap of the probability between positive answer and negative answer will be at large margin. In addition, $\gamma$ is a empirical reward margin to make the training stable. We implement a two-step training strategy. In the first step, which focuses on vision and language alignment, the parameters of the pretrained LLM are frozen. We then train the parameters of vision projector and the components related to Phantom Dimension (MHCA and the functions $f$ and $g$). In the second step, we unfreeze all parameters and train them all at once. We apply PO throughout the first training step only, not to interrupt multimodal LLM's own text generation ability because the positive and negative answers $y^+/y^-$ are mostly generated by closed-source LLVMs instead of instruction fine-tuned self model, which is totally different strategy from RLHF and DPO. For verification, we show the performance degradation in experiment section when using PO in the second training step.

# 4 EXPERIMENTS

**Implementation Details.** To ensure successful reproducibility, we outline four key technical aspects of 🎬 Phantom: (a) the detailed architecture of the backbone multimodal LLMs, vision encoder, and vision projector, (b) the structure of the multi-head cross-attention (MHCA) module in Phantom Dimension, (c) the computing environments and bit quantization configurations, and (d) the procedures for training and inference.

**(a)** We utilize Qwen2 (Yang et al., 2024), Phi-3-mini (Abdin et al., 2024), and InternLM2/2.5 (Cai et al., 2024) as the backbone multimodal LLMs. Specifically, Qwen2-0.5B is configured with $h_q = 14$, $h_{kv} = 2$, a hidden dimension of $d_q = 896$, and 24 layers; InternLM2-1.8B with $h_q = 16$, $h_{kv} = 8$, a hidden dimension of $d_q = 2048$, and 24 layers; Phi-3-mini-3.8B with $h_q = 32$,

Table 3: Detailed comparison for challenging evaluation benchmarks. Sub-benchmark category names in (c), (d), and (g) are represented in Appendix B. For (f), LLaVA-Wilder (Zhang et al., 2024a) is a more advanced challenging evaluation benchmark over LLaVA$^W$ (Liu et al., 2023c).

(a) Comparison with LLLVMs using additional modules and projector: OmniFusion Goncharova et al. (2024), DeepSeek-VL (Lu et al., 2024), MoVA (Kar et al., 2024), Eagle (Shi et al., 2024), CoLLaVO (Lee et al., 2024c), MoAI (Lee et al., 2024d), and Meteor (Lee et al., 2024b)

| Benchmarks | OmniFusion-7B | DeepSeek-VL-7B | MoVA-7B | Eagle-8B | CoLLaVO-7B | MoAI-7B | Meteor-7B | Phantom-7B |
|---|---|---|---|---|---|---|---|---|
| SQA[1] (Lu et al., 2022) | 69.7 | 57.7 | 74.4 | 84.3 | 80.7 | 83.5 | 87.5 | **95.6** |
| MMB (Liu et al., 2023d) | 69.0 | 73.2 | 81.3 | 75.9 | 83.0 | 79.3 | 82.9 | **84.8** |
| MM-Vet (Yu et al., 2023) | 39.4 | 41.5 | - | - | 40.3 | 43.7 | 57.3 | **70.8** |
| MathVista (Lu et al., 2023) | - | - | 44.3 | 52.7 | 57.6 | 56.2 | 53.4 | **70.9** |
| MMStar (Chen et al., 2024d) | - | - | - | - | 42.1 | 48.7 | 45.5 | **57.7** |

(b) Comparison on challenging evaluation benchmarks with more recently released open-source LLLVMs: Cambrian-1 (Tong et al., 2024), LLaVA-OneVision(OV) (Li et al., 2024a), MiniCPM-V-2.6 (Yao et al., 2024), InternVL2 (Chen et al., 2024e), and Qwen2-VL (Wang et al., 2024b), which are trained on larger datasets and with greater computational resources, alongside GPT-4V.

| Benchmarks | Cambrian-1-8B | LLaVA-OV-8B | MiniCPM-V2.6-8B | InternVL2-8B | Qwen2-VL-7B | GPT-4V | Phantom-7B |
|---|---|---|---|---|---|---|---|
| CV-Bench (Tong et al., 2024) | 72.2 | - | - | - | - | 69.1 | **74.9** |
| BLINK (Fu et al., 2024) | 44.9 | 48.2 | - | 50.9 | - | 58.3 | **58.9** |
| MM-Vet (Yu et al., 2023) | 51.7 | 57.5 | 60.0 | 60.0 | 62.0 | 63.6 | **70.8** |
| ChartQA (Masry et al., 2022) | 73.3 | 80.0 | - | 83.3 | 83.0 | 78.5 | **87.8** |
| MathVista (Lu et al., 2023) | 49.0 | - | 60.6 | 58.3 | 58.2 | 69.1 | **70.9** |

(c) MMStar (Chen et al., 2024d)

| LLLVMs | CP | FP | IR | LR | ST | MA | Avg |
|---|---|---|---|---|---|---|---|
| Yi-VL-34B (Young et al., 2024) | 53.2 | 31.2 | 52.0 | 32.4 | 12.4 | 35.2 | 36.1 |
| CogVLM-Chat-17B (Wang et al., 2023) | 66.8 | 36.8 | 49.2 | 31.2 | 23.6 | 11.6 | 36.5 |
| SPHINX-MoE-7B×8 (Gao et al., 2024) | 58.4 | 40.8 | 47.6 | 35.2 | 19.2 | 32.0 | 38.9 |
| InternVL1.2-40B (Chen et al., 2023b) | 67.6 | 43.2 | 61.2 | 47.2 | 24.0 | 19.2 | 43.7 |
| LLaVA-NeXT-34B (Liu et al., 2024a) | 66.4 | 52.0 | 62.4 | 46.0 | 32.4 | 53.6 | 52.1 |
| InternXC2-7B (Dong et al., 2024) | 70.8 | 48.8 | 65.2 | 56.4 | 42.0 | 49.2 | 55.4 |
| GPT-4V (OpenAI, 2023) | 76.6 | 51.4 | 66.6 | 55.8 | 42.6 | 49.8 | 57.1 |
| **Phantom-7B** | 66.0 | 52.8 | 60.0 | 60.8 | 38.4 | 68.4 | 57.7 |

(d) MathVerse (Zhang et al., 2024b)

| LLLVMs | TD | TL | TO | VI | VD | VO | Avg |
|---|---|---|---|---|---|---|---|
| G-LLaVA-7B (Gao et al., 2023) | 20.9 | 20.7 | 21.1 | 17.2 | 16.4 | 9.4 | 16.6 |
| LLaVA-NeXT-13B (Liu et al., 2024a) | 12.8 | 12.0 | 9.9 | 10.7 | 9.7 | 6.3 | 10.3 |
| ShareGPT4V-13B (Chen et al., 2023a) | 16.2 | 16.2 | 6.6 | 15.5 | 13.8 | 3.7 | 13.1 |
| SPHINX-MoE-7B×8 (Gao et al., 2024) | 26.2 | 17.4 | 26.7 | 16.7 | 12.5 | 11.1 | 16.8 |
| InternXC2-7B (Dong et al., 2024) | 22.3 | 17.0 | 16.5 | 15.7 | 16.4 | 11.0 | 16.5 |
| LLaVA-NeXT-34B (Liu et al., 2024a) | 33.8 | 25.5 | 21.3 | 23.5 | 20.3 | 15.7 | 23.8 |
| GPT-4V (OpenAI, 2023) | 54.7 | 41.4 | 48.7 | 34.9 | 34.4 | 31.6 | 39.4 |
| **Phantom-7B** | 47.3 | 45.2 | 45.3 | 42.7 | 41.7 | 43.7 | 41.0 |

(e) MM-Vet-v2 (Yu et al., 2024a)

| LLLVMs | Rec | Gen | OCR | Spat | Know | Seq | Math | Avg |
|---|---|---|---|---|---|---|---|---|
| LLaVA-NeXT34B (Liu et al., 2024a) | 49.3 | 48.9 | 53.2 | 48.3 | 49.6 | 18.5 | 37.3 | 50.9 |
| InternVL-Chat-V1-5 (Chen et al., 2024e) | 52.0 | 48.9 | 51.7 | 49.3 | 47.9 | 37.6 | 17.6 | 51.5 |
| Claude3 Opus (Anthropic, 2024) | 53.5 | 57.6 | 60.5 | 50.0 | 51.0 | 46.1 | 45.6 | 55.8 |
| Qwen-VL-Max (Bai et al., 2023b) | 51.7 | 51.1 | 60.2 | 49.0 | 52.2 | 27.3 | 58.3 | 55.8 |
| Gemini-Pro (Team et al., 2023) | 54.3 | 50.8 | 61.9 | 55.8 | 50.7 | 45.4 | 46.3 | 57.2 |
| **Phantom-7B** | 56.1 | 53.9 | 67.4 | 57.7 | 51.9 | 37.3 | 68.5 | 60.6 |

(f) LLaVA-Wilder

| LLLVMs | Accuracy |
|---|---|
| LLaVA-NeXT-8B (Liu et al., 2024a) | 62.5 |
| LLaVA-NeXT-72B (Liu et al., 2024a) | 71.2 |
| LLaVA-NeXT-110B (Liu et al., 2024a) | 70.5 |
| LLaVA-OV-7B (Li et al., 2024a) | 67.8 |
| LLaVA-OV-72B (Li et al., 2024a) | 72.0 |
| GPT-4V (OpenAI, 2023) | 71.5 |
| **Phantom-7B** | 83.7 |

(g) VisualWebBench Liu et al. (2024b).

| LLLVMs | Website | | | Element | | Action | | Average |
|---|---|---|---|---|---|---|---|---|
| | Cap | QA | OCR | OCR | Grd | Pred | Grd | |
| LLaVA-NeXT-7B (Liu et al., 2024a) | 27.0 | 39.8 | 57.3 | 54.8 | 31.7 | 30.6 | 10.7 | 36.0 |
| LLaVA-NeXT-13B (Liu et al., 2024a) | 26.5 | 44.5 | 52.8 | 56.1 | 31.7 | 48.4 | 15.5 | 39.4 |
| LLaVA-NeXT-34B (Liu et al., 2024a) | 24.3 | 48.2 | 67.1 | 71.9 | 43.1 | 74.0 | 25.2 | 50.5 |
| Gemini-Pro (Team et al., 2023) | 25.0 | 55.5 | 75.1 | 65.4 | 44.3 | 26.7 | 43.7 | 48.0 |
| Claude3 Sonnet (Anthropic, 2024) | 28.9 | 81.8 | 70.3 | 89.2 | 68.8 | 63.4 | 58.3 | 65.8 |
| Claude3 Opus (Anthropic, 2024) | 26.7 | 75.4 | 63.7 | 87.1 | 57.7 | 60.4 | 38.8 | 58.5 |
| GPT-4V (OpenAI, 2023) | 34.5 | 75.0 | 68.8 | 62.8 | 67.5 | 67.6 | 75.7 | 64.6 |
| **Phantom-7B** | 29.0 | 70.2 | 73.8 | 72.3 | 82.8 | 78.6 | 66.9 | 67.7 |

(h) SEED-Bench-2-Plus (Li et al., 2024b)

| LLLVMs | Charts | Maps | Webs | Acc |
|---|---|---|---|---|
| LLaVA-NeXT-7B (Liu et al., 2024a) | 36.4 | 34.0 | 39.9 | 36.8 |
| SPHINX2-13B (Gao et al., 2024) | 41.7 | 41.9 | 60.5 | 48.0 |
| InternXC-7B (Zhang et al., 2023) | 39.9 | 39.0 | 43.0 | 40.6 |
| InternXC2-7B (Dong et al., 2024) | 49.4 | 47.1 | 58.0 | 51.5 |
| SEED-X-13B (Ge et al., 2024b) | 46.9 | 43.3 | 52.6 | 47.1 |
| Gemini-Pro (Team et al., 2023) | 52.1 | 49.4 | 56.8 | 52.8 |
| Claude3 Opus (Anthropic, 2024) | 43.7 | 43.9 | 45.1 | 44.2 |
| GPT-4V (OpenAI, 2023) | 54.8 | 49.4 | 57.2 | 53.8 |
| **Phantom-7B** | 62.5 | 56.4 | 80.5 | 65.5 |

$h_{kv} = 32$, a hidden dimension of $d_q = 3072$, and 32 layers; and InternLM2.5-7B with $h_q = 32$, $h_{kv} = 8$, a hidden dimension of $d_q = 4096$, and 32 layers. For the vision encoder, we employ InternViT-300M (Chen et al., 2023b), which has a hidden dimension of 1024 and 24 layers. The vision projector is designed as MLP that adjusts the hidden dimension from 1024 to match the corresponding multimodal LLM's latent hidden dimension.

**(b)** In each layer, MHCA consists of four linear modules for the query, key, value, and output of the multi-head self-attention operation, where MHCA has similar head dimension for MHSA. For the 0.5B model, the number of parameters required for MHCA module is approximately 1.2M, calculated as $(\frac{896 \text{ (hidden dimension)}}{14 \text{ (number of heads)}})^2 \times 4$ (linear modules) $\times 24$ (layers) $\times 3$ ($qkv$). Similarly, the required parameters for the 1.8B, 3.8B, and 7B models are 4.8M, 3.7M, and 6.2M, respectively. These additional parameters do not significantly impact the overall model size compared with 0.5B, 1.8B, 3.8B, and 7B. Note that, the regularization parameter $\lambda$ during MHSA is set to $\sqrt{2}$.

Table 4: Identifying the effectiveness of 🎬 Phantom by controlling the three factors: Weighted-Average (WA) operation, Phantom Dimension (PD), and Phantom Optimization (PO). If we do not use WA, we then use simple element-wise summation or averaging. In this case, we pick the better performances. Note that, PO-Step1 and -Step2 mean PO is applied in Step1 or Step2.

| | WA | PD | PO-Step1 | PO-Step2 | CV-Bench | BLINK | MMB | SEED-Bench-2-Plus | VisualWebBench | MM-Vet | MM-Vet-v2 | LLaVA-Wilder | MathVista |
|---|---|---|---|---|---|---|---|---|---|---|---|---|---|
| **Phantom-0.5B** | ✗ | ✗ | ✗ | ✗ | 28.2 | 21.4 | 60.7 | 35.7 | 34.7 | 26.6 | 22.0 | 60.8 | 33.8 |
| | ✗ | ✓ | ✗ | ✗ | 29.8 | 21.9 | 62.4 | 39.9 | 37.2 | 27.4 | 22.3 | 64.9 | 36.7 |
| | ✓ | ✓ | ✗ | ✗ | 38.1 | 27.4 | 70.1 | 43.7 | 42.3 | 31.8 | 29.7 | 69.7 | 40.0 |
| | ✓ | ✓ | ✓ | ✗ | **41.5** | **39.3** | **72.7** | **47.9** | **51.8** | **45.7** | **41.5** | **72.2** | **51.7** |
| | ✓ | ✓ | ✗ | ✓ | 36.2 | 36.7 | 68.8 | 40.4 | 47.1 | 39.9 | 36.6 | 67.4 | 48.2 |
| | ✓ | ✓ | ✓ | ✓ | 38.5 | 38.0 | 69.1 | 45.5 | 47.2 | 42.3 | 36.2 | 71.0 | 47.3 |
| | ✗ | ✗ | ✓ | ✗ | 32.0 | 24.2 | 64.2 | 39.0 | 36.4 | 31.2 | 24.1 | 63.4 | 36.8 |
| **Phantom-1.8B** | ✗ | ✗ | ✗ | ✗ | 44.7 | 28.9 | 60.2 | 43.3 | 45.4 | 35.1 | 26.1 | 63.2 | 42.3 |
| | ✗ | ✓ | ✗ | ✗ | 47.0 | 32.6 | 64.7 | 44.9 | 46.5 | 36.0 | 27.4 | 68.7 | 46.4 |
| | ✓ | ✓ | ✗ | ✗ | 52.6 | 35.2 | 69.8 | 50.0 | 53.5 | 41.8 | 32.5 | 71.1 | 49.1 |
| | ✓ | ✓ | ✓ | ✗ | **63.1** | **44.2** | **76.6** | **57.1** | **55.9** | **54.1** | **46.3** | **78.5** | **60.9** |
| | ✓ | ✓ | ✗ | ✓ | 59.9 | 39.9 | 72.2 | 49.7 | 48.4 | 50.5 | 37.0 | 77.1 | 55.9 |
| | ✓ | ✓ | ✓ | ✓ | 59.6 | 40.6 | 73.7 | 54.5 | 55.2 | 53.3 | 41.7 | 76.0 | 58.8 |
| | ✗ | ✗ | ✓ | ✗ | 48.2 | 30.7 | 61.2 | 47.0 | 49.7 | 37.1 | 29.5 | 68.2 | 44.4 |
| **Phantom-3.8B** | ✗ | ✗ | ✗ | ✗ | 63.7 | 34.4 | 62.6 | 42.9 | 45.6 | 38.1 | 32.6 | 73.5 | 45.3 |
| | ✗ | ✓ | ✗ | ✗ | 66.6 | 37.9 | 65.9 | 44.5 | 46.1 | 40.9 | 34.1 | 78.0 | 49.8 |
| | ✓ | ✓ | ✗ | ✗ | 69.1 | 44.1 | 68.9 | 51.8 | 51.8 | 46.9 | 37.0 | 83.4 | 50.5 |
| | ✓ | ✓ | ✓ | ✗ | **73.8** | **51.5** | **80.4** | **61.8** | **59.8** | **54.4** | **48.5** | **85.7** | **60.6** |
| | ✓ | ✓ | ✗ | ✓ | 67.9 | 45.9 | 76.2 | 54.4 | 56.8 | 50.6 | 42.0 | 84.5 | 53.7 |
| | ✓ | ✓ | ✓ | ✓ | 69.2 | 47.8 | 79.2 | 58.6 | 54.8 | 49.9 | 42.9 | 85.0 | 58.1 |
| | ✗ | ✗ | ✓ | ✗ | 68.6 | 37.6 | 65.5 | 47.0 | 46.8 | 41.3 | 36.3 | 76.1 | 49.7 |
| **Phantom-7B** | ✗ | ✗ | ✗ | ✗ | 59.1 | 41.9 | 71.9 | 50.2 | 51.9 | 50.2 | 44.2 | 69.5 | 56.2 |
| | ✗ | ✓ | ✗ | ✗ | 59.8 | 45.9 | 72.5 | 54.2 | 53.6 | 53.7 | 46.3 | 74.4 | 60.9 |
| | ✓ | ✓ | ✗ | ✗ | 69.0 | 47.7 | 81.7 | 59.3 | 57.1 | 62.1 | 53.2 | 77.2 | 64.5 |
| | ✓ | ✓ | ✓ | ✗ | **74.9** | **58.9** | **84.8** | **65.5** | **67.7** | **70.8** | **60.6** | **82.9** | **70.9** |
| | ✓ | ✓ | ✗ | ✓ | 71.7 | 52.2 | 77.9 | 59.1 | 64.1 | 68.2 | 53.1 | 78.7 | 68.1 |
| | ✓ | ✓ | ✓ | ✓ | 70.8 | 54.4 | 82.9 | 60.5 | 66.6 | 69.0 | 54.5 | 82.6 | 68.3 |
| | ✗ | ✗ | ✓ | ✗ | 61.1 | 43.0 | 75.9 | 52.4 | 53.4 | 54.8 | 47.1 | 73.3 | 59.3 |

**(c)** In a computing environment utilizing 8×NVIDIA RTX A6000 48GB GPUs and 8×NVIDIA RTX 3090 24GB GPUs, 🎬 Phantom's training and inference processes take place. To conduct efficient training, each step undergoes a single epoch of training using 8-bit quantization and bfloat16 data format (Kalamkar et al., 2019) for every backbone multimodal LLM. Following bit quantization, we apply QLoRA (Hu et al., 2021; Dettmers et al., 2023) to both vision encoders and backbone multimodal LLMs across all linear layers, using 256 rank and 256 alpha parameters.

**(d)** For Phantom Optimization, we choose equal hyperparameters used in SimPO (Meng et al., 2024): $\beta = 2$ and $\gamma = 0.5$. For training, AdamW optimizer (Loshchilov & Hutter, 2019) is applied, and cosine annealing adjusts the learning rate from 1e-5 to 1e-6 throughout each training step. For multimodal LLM, gradient checkpointing (Sohoni et al., 2019) is employed to manage memory efficiently. A gradient accumulation of 4 leads to batch sizes totaling 128 for each training step, with each step taking roughly two to five days depending on model size. For inference efficiency, 🎬 Phantom is validated using the same quantization level in training, and we make Phantom Dimension cache: $Q_l^*$, $K_l^*$, and $V_l^*$ in each layer to get speedy inference like kv-cache technique, where we use deterministic beam search (Freitag & Al-Onaizan, 2017) ($n = 3$). Memory-efficient scaled dot product attention (SDPA) and FlashAttention2 (Dao et al., 2022; Dao, 2023) accelerates multi-head self-attention (MHSA) computation for Phantom Dimension, benefiting from its hardware-aware ability to mitigate the overhead from the increased latent hidden dimension.

**Validation on 🎬 Phantom.** We present an overview of 🎬 Phantom's vision-language performance in Figure 1-2, and evaluate it on generally used standard evaluation benchmarks as shown in Table 1-2. In the table, LLaVA-OneVision-8B (Li et al., 2024a) uses significant number of image tokens up to 7290 with three training steps on 558K+4M+3.2M datasets. To highlight the benefits of 🎬 Phantom, Table 3 reports performance on more challenging evaluation benchmarks. These results demonstrate that 🎬 Phantom offers a significant advantage on tasks requiring reasoning abilities and densely learned knowledge. Descriptions of the evaluation benchmarks can be found in Appendix C, and 🎬 Phantom's text generation quality is illuminated in Appendix D. In brief, 🎬 Phantom achieves outstanding performance across numerous vision-language tasks, with a large margin over competing LLVMs, despite for a smaller model and fewer instruction tuning samples.

**Ablation Studies.** To better understand the source of this effectiveness, Table 4 and Table 5 presents an ablation study focusing on four key factors: (a) Weighted-Average (WA), (b) Phantom Dimension (PD), (c) Phantom Optimization (PO), and (d) generalization to other LLMs and LLVMs.

Table 5: Application to various pretrained LLMs and LLVMs with Phantom Dimension and Phantom triples. For (c), we empirically find the proper special token that enlarges the latent dimension.

(a) Pretrained LLMs

| LLMs | BLINK | MM-Vet | MathVista |
|---|---|---|---|
| Vicuna1.5-7B | 38.5 | 48.0 | 54.0 |
| +Phantom Dimension | 45.3 | 60.0 | 62.5 |
| +Phantom Triples | 57.1 | 69.5 | 64.0 |
| Gemma-7B | 39.8 | 48.7 | 54.5 |
| +Phantom Dimension | 46.5 | 60.5 | 63.0 |
| +Phantom Triples | 58.0 | 70.1 | 64.7 |
| Mistral-7B | 41.2 | 49.5 | 55.2 |
| +Phantom Dimension | 47.2 | 61.0 | 63.5 |
| +Phantom Triples | 58.5 | 70.3 | 65.3 |
| LLaMA3-8B | 42.7 | 50.0 | 56.0 |
| +Phantom Dimension | 48.9 | 61.8 | 64.2 |
| +Phantom Triples | 59.3 | 70.6 | 66.1 |
| InternLM2.5-7B | 41.9 | 50.2 | 56.2 |
| +Phantom Dimension | 47.7 | 62.1 | 64.5 |
| +Phantom Triples | 58.9 | 70.8 | 70.9 |

(b) Pretrained VLMs

| LLVMs | BLINK | MM-Vet | MathVista |
|---|---|---|---|
| MiniCPM-V2.6-8B | 55.2 | 60.0 | 60.6 |
| +Phantom Dimension | 58.5 | 62.5 | 63.2 |
| +Phantom Triples | 65.1 | 67.0 | 66.5 |
| Cambrian-1-8B | 44.9 | 48.0 | 47.0 |
| +Phantom Dimension | 47.2 | 49.8 | 48.5 |
| +Phantom Triples | 54.0 | 54.5 | 52.0 |
| Molmo-7B | 46.1 | 53.3 | 47.3 |
| +Phantom Dimension | 48.8 | 55.0 | 49.0 |
| +Phantom Triples | 55.0 | 59.1 | 53.0 |
| InternVL2-8B | 50.9 | 54.3 | 58.3 |
| +Phantom Dimension | 53.3 | 56.7 | 60.1 |
| +Phantom Triples | 59.2 | 61.5 | 64.0 |
| LLaVA-OV-7B | 53.0 | 51.9 | 62.3 |
| +Phantom Dimension | 55.6 | 53.8 | 64.5 |
| +Phantom Triples | 60.3 | 62.3 | 68.0 |

(c) Selecting Special Token

| Special Token | BLINK | MM-Vet | MathVista |
|---|---|---|---|
| SOS | **58.9** | **70.8** | **70.9** |
| Image Start | 58.7 | 70.6 | 70.7 |
| Image End | 57.2 | 69.5 | 69.6 |
| Question Start | 57.0 | 69.3 | 69.4 |
| Question End | 56.0 | 68.0 | 68.1 |
| Answer Start | 55.8 | 67.8 | 68.0 |

(d) Larger Pretrained LLVMs

| LLVMs | BLINK | MM-Vet | MathVista |
|---|---|---|---|
| LLaVA-OV-72B | 55.4 | 63.7 | 67.5 |
| +Phantom Dimension | 57.1 | 65.0 | 68.6 |
| +Phantom Triples | **62.0** | **71.8** | **73.8** |
| Qwen2-VL-72B | 60.5 | 73.9 | 69.7 |
| +Phantom Dimension | 62.0 | 75.0 | 70.7 |
| +Phantom Triples | **66.8** | **79.5** | **75.9** |

The results reveal several insights: (1) without Phantom Dimension, training with only positive samples except negative samples (PD:✗, PO: ✗) does not get satisfactory performances more than those with Phantom Dimension (PD:✓, PO: ✗) by using only positive samples, which means Phantom Dimension really works; (2) PD significantly enhances vision-language performance (PD:✓, PO: ✓), as increasing the latent hidden dimension improves the embedding of vision-language knowledge; (3) WA is more effective (WA:✓, PD: ✓) than simple summation or averaging for compressing output features (WA:✗, PD: ✓); (4) PO yields greater performance gains when combined with PD and when applied only during the first training step (PD:✓, PO-Step1: ✓) with a frozen pretrained LLM; (5) other LLMs and LLVMs can be improved by the techniques, (6) even without Phantom Dimension, training with both positive and negative samples (PD:✗, PO: ✗) is more effective than those with only positive samples (PD:✗, PO: ✓) under the absence of Phantom Dimension, which means Phantom Triples really work. Besides, we investigated the effect of replacing the sos token with alternative tokens. We observed using the token that appears earlier in the user question prompt, before the question, is more effective, as shown in Table 5(c). Regarding inference speed, we measured computation time and found only a marginal 10% difference in tokens-per-second between the settings with and without PD. It is attributed to hardware-level operation (SDPA and FlashAttention2 (Dao et al., 2022; Dao, 2023)). Lastly, Appendix E investigate direct comparison of making Phantom Dimension between naively repeating latent dimension and using MHSA and MHCA, and we explore the mixing operation's weights by controlling them which sum up to one or not.

**Discussion and Limitation.** The development of high-performing LLVMs increasingly depends on combining diverse models (Lu et al., 2024; Lee et al., 2024c;d;b; Zong et al., 2024; Shi et al., 2024) and refining existing architectures (Liu et al., 2024c; Lee et al., 2024a), as many aspects of these systems remain unexplored. However, such structural modifications often leads to substantial low-level programming when addressing both development and production-level demands. In response, we will do comprehensive exploration of significantly larger open-source LLVMs, without additional architectural changes. Although there has been a growing trend toward open-source LLVMs, much of the research continues to focus on closed-source LLVMs such as GPT-4V and Gemini-Pro. We either had used GPT-4o-mini and GPT-4o. Therefore, we believe there is untapped potential not only in utilizing the textual outputs of larger open-source LLVMs but also in accessing deeper insights, such as layer-wise features or full parameter sets across layers. Moving forward, we plan to investigate layer-wise distillation methods, which go beyond traditional distillation, to transfer knowledge into models with entirely different architectures using human-understandable language. This direction promises to open up exciting possibilities in a more easier way to develop efficient LLVMs, such as transferring knowledge across heterogeneous structures.

## 5 CONCLUSION

We present an efficient LLVM family 👻 Phantom with significantly enhanced learning capabilities within limited model sizes. By introducing Phantom Optimization (PO) that leverages both autoregressive supervised fine-tuning (SFT) and DPO-like concept, it effectively learns and boosts vision-language performances. Remarkably, despite being smaller than many high-performing LLVMs with larger model sizes, 👻 Phantom demonstrates comparable or even superior performance, making it a promising solution for resource-constrained environments. Our results underscore the power of latent space optimization in boosting both efficiency and performance, offering a pathway toward more efficient LLVMs for various applications.

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

## A PHANTOM TRIPLES

**Question:** ( 2 ^ 2 + 4 ^ 2 + 6 ^ 2 + . . . . . + 14 ^ 2 ) = ?

**Answer:** To find the sum of the squares of the even numbers from 2 to 14, we can calculate each square and then add them together.

Here are the squares of the even numbers from 2 to 14:

2^2 = 4
4^2 = 16
6^2 = 36
8^2 = 64
10^2 = 100
12^2 = 144
14^2 = 196

Now, let's add them together:

4 + 16 + 36 + 64 + 100 + 144 + 196 = 560

So, the sum of the squares of the even numbers from 2 to 14 is 560..

**Rejected:** To find the sum of some numbers, we focus on the pattern of even numbers leading to the total. If we only consider the first few:

2^2 = 4
4^2 = 16
6^2 = 36
8^2 = 64

Now, instead of continuing, let's just quickly check if this looks like it's adding up correctly:

4 + 16 + 36 + 64 = 120, but that might not include everything we need here.

Next, if we think about the complete collection:

2, 4, 6, 8, 10, 12, and possibly 14 could lead to many changes in our approximations.

The sum might generally appear larger but bear in mind we are discussing a specific incremental increase. If we summarize quickly:

The sum of the cited numbers is crucial as it might yield varied totals based on the grouping selected. Hence, the calculation could ultimately reflect values nearest or remote from common assessments. Thus, let's conclude perhaps that it approaches close to 560 without confirming meticulously every addition step comprehensively.

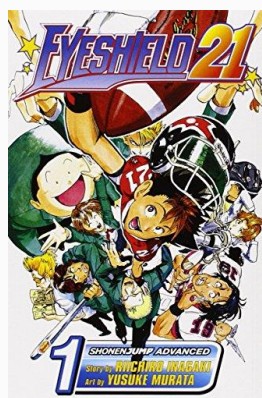

**Question:** What is the title of this book?

**Answer:** Eyeshield 21, Vol. 1

**Rejected:** Gridiron Dreams, Issue 1

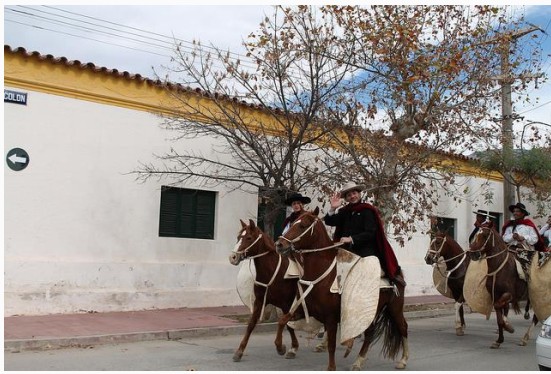

**Question:** What is unique about this scene, especially in a modern city context?

**Answer:** In the image, a group of people dressed in period costumes are riding horses down the street. This is a unique and unusual sight in a modern city context, as horses are not commonly used for transportation or leisure in urban environments nowadays. The presence of individuals dressed in period costumes also adds to the distinctiveness of the scene, as contemporary attire does not typically include these types of outfits. Furthermore, a car is visible in the background, which contrasts with the old-fashioned and historical atmosphere created by the group riding horses down the street. This juxtaposition brings attention to the rarity of such an event in a modern city.

**Rejected:** In the image, a group of people is seen casually strolling through the city while dressed in colorful outfits. This is a striking and unusual sight since such vibrant clothing is rarely spotted in contemporary urban settings. The individuals appear to be enjoying their leisurely walk, creating a lively atmosphere that contrasts with the more subdued tones of the urban environment around them. Additionally, the lack of any vehicles in the immediate vicinity enhances the feeling of tranquility and allows pedestrians to experience the city in a unique way, making this scene feel both modern and timeless.

PAKISTAN

NUTRIENT VALUES USED IN CALCULATIONS (Continued)
(Values per 100 grams "As Purchased" basis)

| Food | Calo-ries | Pro-tein Gms | Fat Gms. | Ca Mg | Fe Mg | Vit. A I. U. | Thia-mine Mg | Ribo-flavin Mg | Niacin Mg | Vit. C Mg | Ref. Item 1/ |
|---|---|---|---|---|---|---|---|---|---|---|---|
| Carrots | 37 | 1.1 | 0.3 | 34 | 0.7 | 10,560 | 0.05 | 0.05 | 0.5 | 5 | A 120 |
| Peas, fresh | 45 | 3.0 | 0.2 | 10 | 0.9 | 310 | 0.15 | 0.07 | 1.2 | 12 | A 160 |
| Cabbage | 17 | 1.0 | 0.1 | 34 | 0.4 | 60 | 0.04 | 0.04 | 0.2 | 36 | A 118 |
| Spinach | 16 | 1.9 | 0.2 | 66 | 2.5 | 7,720 | 0.09 | 0.16 | 0.5 | 48 | A 176 |
| Coriander,Fresh | 32 | 2.2 | 0.4 | 152 | 5.3 | 5,350 | 0.09 | 0.11 | 0.9 | 75 | A 131 |
| Karela (Momordica charantia) | – | – | – | – | – | (?) | – | – | – | 140 | D |
| Eggplant | 23 | 1.1 | 0.2 | 14 | 0.4 | 30 | 0.04 | 0.05 | 0.6 | 5 | A 136 |
| Potato | 70 | 1.7 | 0.1 | 9 | 0.6 | – | 0.09 | 0.03 | 1 | 14 | A 163 |
| Onion, dry | 42 | 1.3 | 0.2 | 30 | 0.5 | 50 | 0.03 | 0.04 | 0.2 | 8 | A 154 |
| Onion, green | 24 | 1.1 | 0.2 | 27 | 0.6 | 20 | 0.06 | 0.03 | 0.3 | 9 | A 145D |
| Turnip | 28 | 1.0 | 0.2 | 35 | 0.4 | – | 0.04 | 0.06 | 0.4 | 24 | A 187 |
| Cauliflower | 14 | 1.4 | 0.1 | 13 | 0.6 | 50 | 0.06 | 0.06 | 0.3 | 39 | A 122 |
| Veg. Marrow | 20 | 0.7 | – | – | 0.7 | – | – | – | – | 18 | D |
| Tomato, fresh | 18 | 0.9 | 0.3 | 10 | 0.5 | 970 | 0.05 | 0.04 | 0.04 | 20 | A 184 |
| Tomato, can | 98 | 2 | 0.4 | 12 | 0.8 | 1,880 | 0.09 | 0.07 | 2.2 | 11 | A 361 |
| Orange, malta | 32 | 0.6 | 0.1 | 24 | 0.3 | 140 | 0.06 | 0.02 | 0.1 | 35 | A 71 |
| Guava | 58 | 0.8 | 0.5 | 13 | 0.7 | 160 | 0.05 | 0.03 | 0.8 | 246 | A 53 |
| Papaya | 27 | 0.4 | 0.1 | 14 | 0.2 | 1,190 | 0.02 | 0.03 | 0.2 | 38 | A 77 |
| Banana | 59 | 0.8 | 0.1 | 5 | 0.4 | 290 | 0.03 | .03 | 0.5 | 7 | A 41 |
| Tamarind | 115 | 1.3 | 0.3 | 36 | 0.3 | 10 | 0.16 | 0.07 | 0.6 | 1 | A 96 |
| Raisins | 268 | 2.3 | 0.5 | 78 | 3.3 | 50 | 0.15 | 0.08 | 0.5 | – | C 553 |
| Garlic | 84 | 4.0 | 0.2 | 37 | 0.9 | – | 0.19 | 0.07 | 0.4 | 13 | A 139 |
| Chillies, dry | 245 | 14.8 | 7 | 15 | 1.5 | 575 | – | – | – | 50 | D., E |
| Spices, mix | 325 | 11.8 | 11.4 | 53 | 21 | 822 | – | – | – | – | D |

1/ References: A. Composition of foods used in Far Eastern countries. USDA Handbook #34, 1952
B. Food Composition Tables - minerals and vitamins for international use. FAO United Nations, 1954.
C. Composition of Foods - raw, processed, prepared. USDA Handbook #8, 1950.
D. Major M. Iqbal, GHQ Science Laboratories, Chuklala, Pakistan.
E. Analyses of Medical Nutrition Laboratory, Denver, Colorado.

**Question:** How many calories per 100 grams of Raisins

**Answer:** 268

**Rejected:** Raisins contain approximately 300 calories per 100 grams.

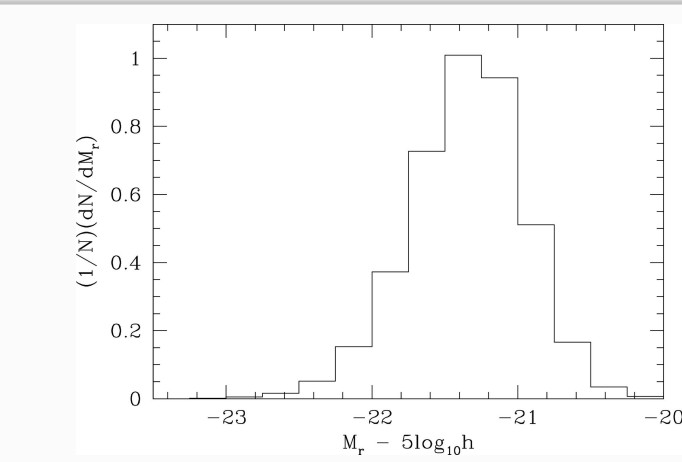

**Question:** What can be inferred about the luminosity of the galaxies surveyed in this graph?

**Answer:** The graph shows a clear peak within a specific magnitude range (-22 to -21), indicating that there is a concentration of galaxies with luminosities corresponding to that absolute magnitude range.

**Rejected:** The graph displays various values across different magnitudes, which may suggest that there is a diversification in the luminosity of the observed galaxies, but without a clear trend or dominant cluster within any specific range.

1134
1135
1136
1137
1138
1139
1140
1141
1142
1143
1144
1145
1146
1147
1148
1149
1150
1151
1152
1153
1154
1155
1156
1157
1158
1159
1160
1161
1162
1163
1164
1165
1166
1167
1168
1169
1170
1171
1172
1173
1174
1175
1176
1177
1178
1179
1180
1181
1182
1183
1184
1185
1186
1187

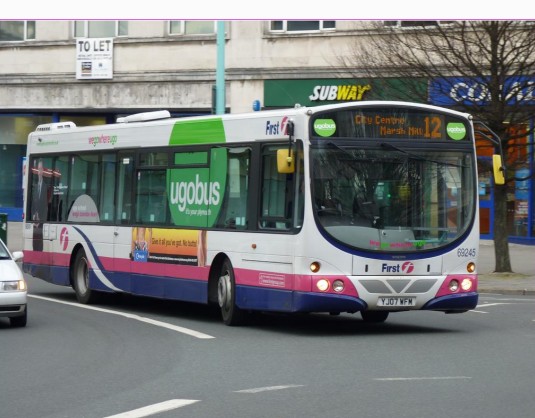

**Question:** Summarize the visual content of the image.

**Answer:** A bus driving down the street is heading towards City Centre.

**Rejected:** A bicycle parked near a café is surrounded by pedestrians.

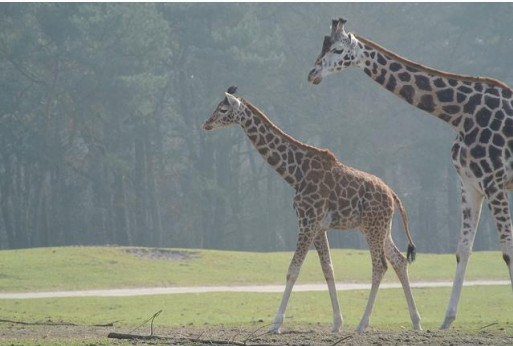

**Question:** Please illustrate the image through your words.

**Answer:** The image features two giraffes in a natural habitat that resembles a savannah or open woodland. The giraffes are positioned side by side, moving from left to right across the frame. The foremost giraffe is younger and smaller, with a lighter and more uniform pattern of spots. The other, presumably an adult, has darker, more irregular spots and is slightly ahead of the younger one. They both have long necks stretched upwards, and their bodies are slender and tall, typical of their species. The background is blurred but shows a landscape with sparse trees and grass under a clear sky. The colors are soft, with earthy tones dominating the scene. There's a sense of calmness and tranquility.

**Rejected:** The image depicts a group of animals in a dense forest setting. The creatures appear to be moving together, with one noticeably larger than the others. The larger animal has a distinct pattern on its body, while the smaller one seems to be following closely behind. The background is filled with thick foliage and shadows, creating a sense of mystery. The colors are rich and vibrant, with deep greens and browns dominating the scene. There's an atmosphere of liveliness and energy, suggesting that the animals are engaged in some form of activity.

## B    GATHERED CONFIGURATION OF VISUAL INSTRUCTION TUNING SAMPLES

To cover a broad range of capabilities, we compile 2.8M visual instruction tuning samples across multiple datasets, encompassing various domains such as fundamental image understanding, real-world common-sense knowledge, non-object visual concepts (e.g., documents, charts, diagrams, symbols, and signs), and general mathematical problems. Our dataset collection includes basic image understanding samples from ShareGPT4o-Images (57K) (Erfei Cui, 2024), ShareGPT4V (755K) (Chen et al., 2023a), ALLaVA-VFLAN/Text (548K) (Chen et al., 2024b), and MiniGemini (27K) (Li et al., 2024d) targeting tasks for DocVQA (Mathew et al., 2021), ChartQA (Masry et al., 2022), DVQA (Kafle et al., 2018), and AI2D (Kembhavi et al., 2016). Additionally, we collect LLaVA-HD (116K) (Zhang et al., 2024c) for Science and Mathematical Reasoning (SMR), supporting ArXivQA (Li et al., 2024c) and TextbookQA (Kembhavi et al., 2017), and we further integrate document understanding samples from mPLUG-DocOwl1.5-Downstream/Reasoning (599K) (Hu et al., 2024a) and general mathematical problems from GLLaVA (177K) (Gao et al., 2023), MathVision (3K) (Wang et al., 2024a), MathInstruct (262K) (Yue et al., 2023), and MathPlus (304K) (Yue et al., 2024).

## C    DESCRIPTION OF EVALUATION BENCHMARKS

- **SQA-IMG (SQA[I])** (Lu et al., 2022) is part of the broader ScienceQA (SQA) dataset, which aims to improve reasoning and interpretability in AI systems through science-based question answering. This dataset covers a wide range of science disciplines, featuring 26 different topics in natural, social, and language sciences, all accompanied by annotated answers, lectures, and explanations. SQA-IMG includes image-related samples, amounting to 10,332 question-answer pairs.

- **AI2D** (Kembhavi et al., 2016) or AI2 Diagrams, addresses diagram interpretation and reasoning challenges, focusing on syntactic parsing and semantic understanding. It supports research into diagram structure and element relationships, critical for tasks like diagram-based question answering. This collection includes over 5,000 diagrams from elementary science topics, along with over 15,000 multiple-choice questions.

- **ChartQA** (Masry et al., 2022) develops to challenge and improve question answering systems that deal with data visualizations like bar charts, line charts, and pie charts. This dataset tests systems on questions requiring arithmetic and logical reasoning and includes both human-generated and machine-created question-answer pairs. It comprises 32,719 samples in total.

- **SEED-IMG (SEED[I])** (Li et al., 2023a), a subset of SEED-Bench, evaluates the generative comprehension skills of multimodal large language models (MLLMs) with a focus on spatial and temporal understanding. It offers several subsets mapped to 12 evaluation dimensions across image and video modalities, with SEED-IMG specifically concentrating on images.

- **SEED-Bench-2-Plus** (Li et al., 2024b) evaluates multimodal large language models in their ability to understand text-rich visual content, common in real-world settings like charts, maps, and website interfaces. This benchmark specifically measures how effectively MLLMs can interpret these complex, text-rich scenarios that require simultaneous comprehension of visual and textual information. The benchmark is divided into three main categories—Charts, Maps, and Webs, and further subdivided into 63 unique data types with 2.3k multiple-choice questions.

- **POPE** (Li et al., 2023b) introduces a method to systematically assess the tendency of LLVMs to falsely generate nonexistent objects in images. This method turns the evaluation into a binary classification task using polling questions, providing a fair and adaptable approach.

- **HallusionBench (HallB)** (Liu et al., 2023a) is crafted to evaluate and explore visual illusions and knowledge hallucinations in large language and vision models (LLVMs). This benchmark uses carefully crafted example pairs to identify model failures, featuring diverse visual-question pairs including subsets focused on illusions, math, charts, tables, maps, and OCR. It includes 346 images and 1,129 questions.

- **MME** (Fu et al., 2023) serves as a comprehensive evaluation framework for Multimodal Large Language Models (MLLMs), focusing on various perception and cognition tasks through 14 sub-tasks like coarse and fine-grained recognition, OCR, and commonsense reasoning. This benchmark aims to address existing evaluation gaps and ensures a thorough testing environment for MLLMs.

- **MathVista** (Lu et al., 2023) is an extensive benchmark designed to test visual-based mathematical reasoning in AI models. It integrates visual understanding in evaluating models' abilities to solve math problems that involve visuals. The dataset consists of three subsets: IQTest, FunctionQA, and PaperQA, totaling 6,141 examples.

- **MMB, MMB-Chinese (MMB$^{CN}$)** (Liu et al., 2023d) aims to establish a robust evaluation standard for vision language models by covering a broad spectrum of necessary multimodal comprehension skills (20 fine-grained abilities) in both English and Chinese. This benchmark consists of 3,217 questions gathered from various sources to challenge different facets of LLVMs.

- **MM-Vet** (Yu et al., 2023) is designed to systematically evaluate LMMs on complex tasks requiring multiple vision language (VL) capabilities. It tests recognition, knowledge, OCR, spatial awareness, language generation, and math, integrating these abilities into 16 different task combinations. The dataset includes 200 images and 218 questions, each requiring the integration of multiple capabilities.

- **MM-Vet-v2** (Yu et al., 2024a) evaluates a wide range of integrated abilities in large multimodal models, such as Recognition, Knowledge, Optical Character Recognition (OCR), Spatial Awareness, Language Generation, Math, and Image-Text Sequence Understanding. This version builds upon the original MM-Vet benchmark by adding tasks that involve comprehending sequential information from both images and text, which is essential for real-world scenarios. MM-Vet-v2 places a strong focus on assessing the model's capacity to interpret and reason through intricate image-text sequences. The benchmark includes 517 evaluation samples, a notable increase from the 217 samples in the original MM-Vet.

- **LLaVA Bench in the Wild(er) (LLaVA$^W$ and LLaVA-Wilder )** (Liu et al., 2023c; Zhang et al., 2024a) assesses large multimodal models (LMM) on complex tasks and new domains through a collection of 24 images with 60 questions for 'wild' and its more advanced version of 'wilder'. This dataset features diverse settings, including indoor, outdoor, artworks, and memes, with each image accompanied by detailed descriptions and curated questions.

- **MMStar** (Chen et al., 2024d) is crafted to precisely evaluate the true multimodal capabilities of LLVMs by ensuring that each sample critically relies on visual content for accurate answers while minimizing data leakage. It comprises 1,500 meticulously selected samples and is organized into six primary sub-benchmarks as follows:

  - **Coarse perception (CP)**, which pertains to the ability to grasp and interpret the overarching features and themes of an image without focusing on minute details,

  - **Fine-grained perception (FP)**, which denotes a detailed level of image comprehension that emphasizes the intricate and nuanced aspects of visual content,

  - **Instance reasoning (IR)**, which encompasses advanced cognitive abilities aimed at understanding and interpreting individual and collective object attributes and their interrelations within an image,

  - **Logical reasoning (LR)**, which involves a sophisticated framework of cognitive processes designed to interpret, deduce, and infer conclusions from visual content through a structured approach to logic and reasoning,

  - **Science & technology (ST)**, which includes a comprehensive framework for the application and integration of knowledge across a wide range of scientific and technological domains,

  - **Math (MA)**, which is a fundamental pillar of logical and analytical reasoning and includes a broad spectrum of skills essential for understanding, applying, and interpreting quantitative and spatial information.

- **MathVerse** (Zhang et al., 2024b) assesses the capabilities of Multi-modal Large Language Models (MLLMs) in visual mathematical reasoning, particularly their ability to understand

visual diagrams and mathematical expressions. This dataset is categorized into three primary areas: plane geometry, solid geometry, and functions, and further detailed into twelve types like length and area, encompassing 2,612 visual mathematical challenges.

To investigate how MLLMs process visual diagrams in mathematical reasoning, the creators of MathVerse developed six distinct versions of each problem, each version presenting different levels of multi-modal information. They initially established three specific classifications for the text content within the problems:

- *Descriptive Information*, which includes content that is directly visible and explicitly depicted in the diagrams,
- *Implicit Property*, which encompasses details that demand a more advanced visual perception yet less mathematical knowledge to interpret from the diagram,
- *Essential Condition*, which pertains to crucial numerical or algebraic data necessary for solving the problem that cannot be inferred solely from the visual diagram.

Based on these categories, to thoroughly assess the true visual understanding capabilities of MLLMs and their utility in multi-modal mathematical contexts, the researchers created six versions or sub-benchmarks of each problem in MathVerse, described as follows:

- **Text dominant (TD)** version, which preserves all textual elements, including the three textual categories and the main question, prompting MLLMs to primarily depend on textual information.
- **Text lite (TL)** version reduces the *Descriptive Information* from the Text dominant version, promoting reliance on the diagram for elementary data.
- **Text only (TO)** version removes the visual elements entirely, focusing on textual content to discern where MLLMs predominantly derive contextual information for problem solving.
- **Vision intensive (VI)** further excludes *Implicit Property* from the Text lite version, urging MLLMs to intensify their visual analysis to gather essential cues for mathematical reasoning.
- **Vision dominant (VD)**, evolving from the Text lite version, omits *Essential Condition* from the textual information and instead visually annotates these details in diagrams, compelling MLLMs to identify and accurately link these essential conditions solely through visual examination.
- **Vision only (VO)** eliminates all textual descriptions, presenting the problem exclusively through visual means and challenging MLLMs to decode and identify mathematical queries purely based on visual data, serving as the ultimate test of their visual reasoning skills in mathematics.

- **VisualWebBench** (Liu et al., 2024b) assesses the capabilities of multimodal large language models (MLLMs) specifically in the web domain. It is designed to address the lack of a comprehensive benchmark that evaluates the unique characteristics of web pages and measures fine-grained abilities such as OCR, understanding, and grounding (Grd) in text-rich and interactive web environments. It covers a wide range of domains, including science, travel, sports, engineering, and government, and tasks such as captioning (Cap), WebQA (QA), heading OCR, element grounding (Grd), and action prediction (Pred), containing a total of 1,534 instances.

- **CV-Bench** (Tong et al., 2024) is designed for vision-focused evaluation in multimodal large language models. This benchmark aims to fill the gaps in traditional benchmarks, which often fall short in thoroughly assessing visual grounding in real-world contexts. CV-Bench assesses the model's abilities in both 2D and 3D visual tasks using natural language questions. The evaluation is split into 2D tasks (such as spatial relationships and object counting) and 3D tasks (like depth order and relative distance), providing a well-rounded test of the model's visual comprehension with 2,638 carefully inspected examples.

- **BLINK** (Fu et al., 2024) is created to assess the visual perception capabilities of multimodal large language models. It features 14 key visual perception tasks, which are based on traditional computer vision challenges but restructured into 3,807 multiple-choice questions that involve one or more images. These tasks address difficulties such as relative depth estimation, visual correspondence, forensic detection, and multi-view reasoning.

Additionally, we will continue to explore more challenging evaluation benchmarks to uncover previously unaddressed issues such as Yu et al. (2024b), advancing 👻 Phantom through ongoing technical development. By leveraging a wide range of methods established over the years (Lee, 2020; Lee et al., 2021; Kim et al., 2021; Lee et al., 2022; Kim et al., 2023b; Lee et al., 2023; Kim et al., 2023a;c; Park et al., 2024c;b; Kim et al., 2024), we aim to drive innovative breakthroughs across both general and specialized tasks.

# D 👻 PHANTOM GENERATION QUALITY

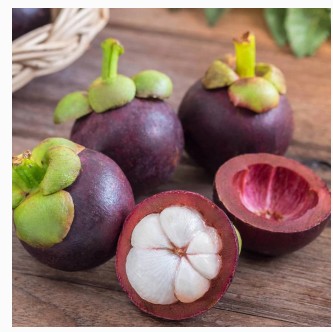

**User:** What type of fruit is this?

**Phantom:** The fruit in the image is a mangosteen. It is characterized by its deep purple or red outer shell and segmented white flesh inside. The mangosteen is known for its sweet and tangy flavor, often compared to a combination of strawberries and grapes. It is a popular fruit in Southeast Asia and is considered a luxury fruit due to its unique taste and the effort required to open the tough outer shell.

**User:** Write a fairy tale based on this painting.

**Phantom:** Once upon a time, in a lush, enchanted forest, there lived three unlikely friends: Bruno the Bear, Whiskers the Cat, and Thumper the Rabbit. They shared a love for tea parties under the shade of a magnificent oak tree.

One sunny afternoon, as they gathered around their favorite tree stump, Bruno brought out a plate of freshly baked cookies. Whiskers, with her keen sense of taste, brewed a pot of the finest herbal tea. Thumper, known for his playful antics, brought along a basket of wildflowers to decorate the table.

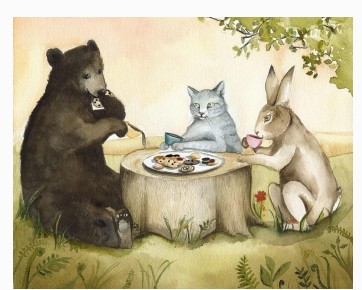

As they sipped their tea and nibbled on cookies, they noticed a mysterious map tucked under the plate. It showed a path leading to a legendary treasure hidden deep within the forest. Excited by the adventure, they decided to set out on a journey to find the treasure.

Bruno, with his strength, carried the map; Whiskers, with her sharp eyes, navigated the way; and Thumper, with his speed, led them through the dense undergrowth. They encountered challenges along the way, from crossing babbling brooks to outsmarting mischievous forest creatures.

After many days of travel, they reached a clearing where a magnificent tree stood. At its base, they found a chest filled with sparkling jewels and ancient coins. But the real treasure was their friendship, which shone brighter than any gem.

From that day on, Bruno, Whiskers, and Thumper continued their tea parties under the oak tree, sharing their adventures and the joy of their friendship. The forest was a better place because of their unity, and they all lived happily ever after.

1404
1405
1406
1407
1408
1409
1410
1411
1412
1413
1414
1415
1416
1417
1418
1419
1420

**User:** I am getting one funghi pizza and one prosciutto pizza. How much should I pay in total?

**Phantom:** To calculate the total cost for one Funghi pizza and one Prosciutto pizza, we need to look at the prices on the menu:

- Funghi pizza costs 6.90€.
- Prosciutto pizza costs 7.90€.

Adding these together:

6.90€ + 7.90€ = 14.80€

So, you should pay a total of 14.80€ for one Funghi pizza and one Prosciutto pizza.

1421
1422
1423
1424
1425
1426
1427
1428
1429
1430
1431
1432
1433
1434
1435
1436
1437
1438
1439
1440

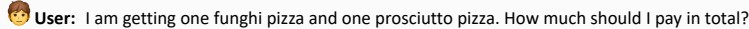

| | JAN | FEB | MAR | APR | MAY | JUN | Total | Average |
|---|---|---|---|---|---|---|---|---|
| **INCOME** | | | | | | | | |
| Salary | $5,000 | $5,000 | $5,000 | $5,000 | $5,000 | $5,000 | $30,000 | $5,000 |
| Business | $4,000 | $4,200 | $3,800 | $3,400 | $4,000 | $3,600 | $23,000 | $3,833 |
| Dividends | 0 | 0 | 0 | 0 | 0 | 0 | $0 | $0 |
| Interest Income | $50 | $50 | $50 | $50 | $50 | $50 | $300 | $50 |
| Others | $0 | $0 | $20 | $0 | $30 | $0 | $50 | $8 |
| **Total** | $9,050 | $9,250 | $8,870 | $8,450 | $9,080 | $8,650 | $53,350 | $8,892 |

**User:** What percentage does salary contribute to total income?

**Phantom:** To determine the percentage contribution of salary to total income, we need to calculate the total salary over the six months and then find its percentage of the total income.

1. **Calculate Total Salary:**
- Salary for each month is $5,000.
- Total salary over six months = $5,000 * 6 = $30,000.

2. **Calculate Total Income:**
- Total income over six months = $53,350.

3. **Calculate the Percentage Contribution:**
- Percentage contribution of salary = (Total Salary / Total Income) * 100
- Percentage contribution of salary = ($30,000 / $53,350) * 100

4. **Perform the Calculation:**
- Percentage contribution of salary = (30,000 / 53,350) * 100 ≈ 56.14%.

Therefore, the salary contributes approximately 56.14% to the total income.

1441
1442
1443
1444
1445
1446
1447
1448
1449
1450
1451
1452
1453
1454
1455
1456
1457

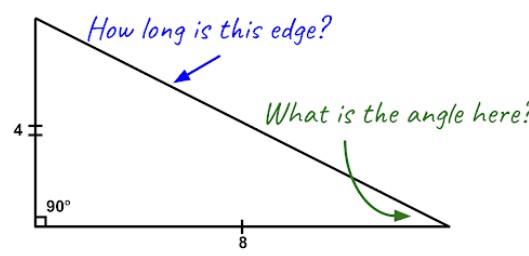

**User:** Answer the question in the following figure. Provide intermediate steps.

**Phantom:** To determine the length of the edge labeled "How long is this edge?" and the angle labeled "What is the angle here?" in the given right triangle, we can use trigonometric functions.

1. **Determining the Length of the Edge:**
   - The triangle is a right triangle with one angle measuring $90°$.
   - The given side lengths are 4 (opposite side) and 8 (adjacent side) to the angle in question.

Using the tangent function, which is the ratio of the opposite side to the adjacent side:

$$\tan(\theta) = \frac{\text{opposite}}{\text{adjacent}} = \frac{4}{8} = \frac{1}{2}$$

To find the angle $\theta$ :

$$\theta = \tan^{-1}\left(\frac{1}{2}\right)$$

Using a calculator:

$$\theta \approx 26.57°$$

2. **Determining the Length of the Edge:**
   - To find the hypotenuse (the edge labeled "How long is this edge?"), use the Pythagorean theorem:

$$\text{Hypotenuse}^2 = \text{opposite}^2 + \text{adjacent}^2$$
$$\text{Hypotenuse}^2 = 4^2 + 8^2 = 16 + 64 = 80$$
$$\text{Hypotenuse} = \sqrt{80} = \sqrt{16 \times 5} = 4\sqrt{5}$$

Therefore, the length of the hypotenuse is $4\sqrt{5}$.

3. **Determining the Angle:**
   - The angle opposite the side of length 4 is:

$$\theta = \tan^{-1}\left(\frac{4}{8}\right) = \tan^{-1}\left(\frac{1}{2}\right) \approx 26.57°$$

The angle adjacent to the side of length 8 is:

$$90° - 26.57° = 63.43°$$

Thus, the length of the edge is $4\sqrt{5}$ and the angle is $26.57°$.

👤 **User:** Analyze the images and decide what the next steps should be.

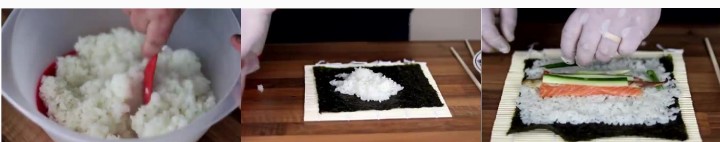

👻 **Phantom:** The images depict the process of making sushi. Based on the sequence:

1. **Rice Preparation**: The first image shows rice being mashed with a spoon.
2. **Sushi Formation**: The second image shows the rice being shaped into a mound on a sheet of nori (seaweed).
3. **Adding Fillings**: The third image shows the addition of fillings like cucumber and salmon.

The next steps should be:

4. **Rolling the Sushi**: After adding the fillings, the sushi should be rolled using the nori sheet as a base. This involves using a rolling mat to ensure the sushi is neatly rolled.

5. **Cutting the Sushi**: Once rolled, the sushi should be cut into bite-sized pieces using a sharp knife.

These steps will complete the sushi-making process.

👤 **User:** What breed is this dog and how should I properly care for it?

👻 **Phantom:** The dog in the image is a Poodle. Poodles are a popular breed known for their intelligence, friendly demeanor, and low-shedding coats. Here's how to care for a Poodle:

1. **Grooming**: Due to their curly fur, Poodles require frequent grooming. Brush their coat daily to prevent tangles and mats. Schedule haircuts every 6-8 weeks to keep their coat healthy and manageable.

2. **Exercise**: Poodles are active dogs and need regular physical activity. Ensure they get at least 30 minutes of daily exercise, such as walks or play sessions.

3. **Training**: Being an intelligent breed, Poodles thrive with training. Early socialization and obedience training can help them become well-behaved and obedient dogs.

4. **Health**: Keep up with regular vet visits for preventive care. Poodles are generally healthy, but they can be prone to certain conditions like Addison's disease. Be aware of these potential health issues.

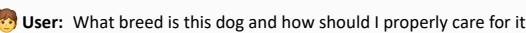

5. **Nutrition**: Feed your Poodle a high-quality dog food suitable for their age, size, and activity level. Consult with your veterinarian for specific dietary needs.

6. **Mental Stimulation**: Poodles enjoy mental challenges. Provide them with interactive toys or training sessions to keep them mentally engaged.

7. **Hygiene**: Ensure your Poodle is clean by bathing them every 4-6 weeks. Brush their teeth regularly to prevent dental issues.

By following these care tips, you can help your Poodle stay healthy, happy, and well-groomed.

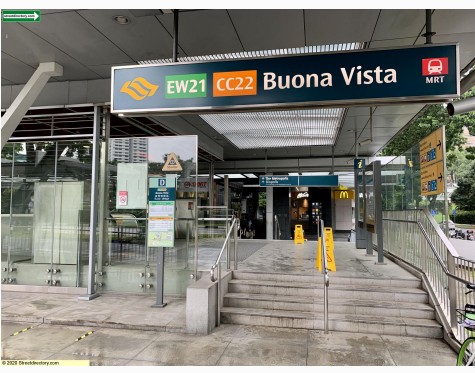

🧑 **User:** Could you provide detailed directions on how to reach the nearest McDonald's from this MRT station?

👻 **Phantom:** To reach the nearest McDonald's from Buona Vista MRT station, exit through the main entrance and proceed straight. As you walk, you'll notice a McDonald's sign on your right. Descend the steps and keep going straight until you see the McDonald's sign. The McDonald's will be on your right side.

# E   DISCUSSION FOR PHANTOM DIMENSION

Comparison of methods for enlarging latent dimensions.

| Enlarging Latent Dims | BLINK | MM-Vet | MathVista |
|---|---|---|---|
| Just Repeat | 44.5 | 55.3 | 59.0 |
| MHCA+MHSA | **58.9** | **70.8** | **70.9** |

Performance comparison of mixing strategies.

| Mixing | BLINK | MM-Vet | MathVista |
|---|---|---|---|
| Mixing (the sum of weights are not limited) | 55.3 | 64.7 | 67.1 |
| Mixing (the sum of weights are one) | **58.9** | **70.8** | **70.9** |

