# OpenReview forum: "Phantom of Latent for Large Language and Vision Models"
_ICLR.cc/2025/Conference — Submitted to ICLR 2025_

### Official Review · Reviewer_Vpt7 · 2024-11-02

**Soundness:** 3
**Presentation:** 2
**Contribution:** 3
**Rating:** 6
**Confidence:** 2

**Summary:**

The paper proposes Phantom, a new family of Large Language and Vision Models (LLVMs) that achieve comparable or superior performance to larger models while maintaining a smaller model size. They achieve this through a new cross attention architecture "Phantom Dimension", which temporarily expands the latent hidden dimension during multi-head self-attention, and "Phantom Optimization" (PO), a training strategy that minimizes the generation of incorrect and ambiguous answers. They claim Phantom outperforms existing open- and closed-source LLVMs in its size category across multiple benchmarks.

**Strengths:**

- Introduced techniques are novel and offer a potentially valuable new direction for efficient LLVM design.
- The reported results demonstrate impressive performance compared to other LLVMs, especially in the lower parameter regime (0.5B, 1.8B), suggesting effectiveness.
- The ablation study provides insights into the contribution of each proposed component, although some aspects could be further explored.

**Weaknesses:**

## Major

The paper's writing quality hinders clear communication of the results and techniques, making evaluation difficult.  For example, the abstract states:  "...LLMs either have further increased their sizes, reaching 26B, 34B, and even 80B parameters." This sentence is awkwardly phrased. The authors should prioritize improving the writing throughout the paper for clarity.  Using an LLM grammar checker would be beneficial.

The explanation of the proposed "Phantom Dimension" attention mechanism on page 5 lacks clarity.  A clearer presentation would first describe the attention mechanism used in the baseline architecture and then highlight the novel aspects introduced by Phantom. Figure 3 does not adequately illustrate these distinctions.

The paper introduces multiple novel elements: a new dataset, a potentially novel attention mechanism, and a new loss function. However, the ablation study omits the dataset. This omission prevents a proper assessment of whether the state-of-the-art results stem from the data itself or the specific combination of the Phantom architecture and optimization techniques.

The paper lacks a comparison to Malmo, a relevant work found at: https://arxiv.org/abs/2409.17146. This comparison is crucial for contextualizing the contributions.

## Minor

The term "projectors" is used without clear definition.  If these are Multilayer Perceptron (MLP) layers, that term should be used for clarity and consistency.

**Questions:**

- Do you plan to share weights, code and data for training Phantom?

---

> ### Author Response · Authors · 2024-11-17
> **Rebuttals**
>
> **Q1. The paper's writing quality hinders clear communication of the results and techniques, making evaluation difficult. For example, the abstract states: "...LLMs either have further increased their sizes, reaching 26B, 34B, and even 80B parameters." This sentence is awkwardly phrased. The authors should prioritize improving the writing throughout the paper for clarity. Using an LLM grammar checker would be beneficial.**
>
> A1. We've edited it. You can see the updated paper.
>
> ---
>
> **Q2. The explanation of the proposed "Phantom Dimension" attention mechanism on page 5 lacks clarity. A clearer presentation would first describe the attention mechanism used in the baseline architecture and then highlight the novel aspects introduced by Phantom. Figure 3 does not adequately illustrate these distinctions.**
>
> A2. Based on your comment, we've added the algorithm on the updated paper.
>
> **Algorithm 1**: A Transformer Decoder Block with Phantom Dimension
>
> 1. **Input**: $X = [x_{sos}, x_{prompt}] $
> 2. $ Q_l, K_l, V_l \leftarrow \text{LinearQKV}(X) $
> 3. $ Q_l^*, K_l^*, V_l^* \leftarrow Q_l[0], K_l[0], V_l[0] $ {Extracting features for the place of the 'sos' token}
> 4. $ Q_l^*, K_l^*, V_l^* \leftarrow \text{MHCA}(Q_l, Q_l^*), \text{MHCA}(K_l, K_l^*), \text{MHCA}(V_l, V_l^*) $
> 5. $[ \bar{O}_l, \tilde{O}_l ] \leftarrow \text{MHSA}(q = [Q_l, Q_l^*], k = [K_l, K_l^*], v = [V_l, V_l^*])$
> 6. $ \bar{w}, \tilde{w} \leftarrow e^{f(\bar{O}_l)} / (e^{f(\bar{O}_l)} + e^{g(\tilde{O}_l)}), e^{g(\tilde{O}_l)} / (e^{f(\bar{O}_l)} + e^{g(\tilde{O}_l)}) $
> 7. $O_l \leftarrow \bar{w} \odot \bar{O}_l + \tilde{w} \odot \tilde{O}_l$
> 8. $X\leftarrow (\text{FFN}+\text{Add-Norm})(O_l) $
> 9. **Return**: $X $
>
> ---
>
> **Q3.  The ablation study omits the dataset. This omission prevents a proper assessment of whether the state-of-the-art results stem from the data itself or the specific combination of the Phantom architecture and optimization techniques.**
>
> A3. Disagree. We would like to clarify our Table4. For example, PD: X, PO: X means normal visual instruction tuning without Phantom Dimension. In addition, PD: X, PO: O means visual instruction tuning with negative samples but without Phandom Dimension. Therefore, we already experimented what you pointed out. It would be really helpful if you suggest what Table4 didn't show. Nonetheless, we can show some other pretrained LLMs or pretrained LLVMs + Phantom.
>
> | LLMs             | BLINK | MM-Vet | MathVista |
> |-------------------|-------|--------|-----------|
> | Vicuna1.5-7B     | 38.5  | 48.0   | 54.0      |
> | +Phantom Dimension | 45.3  | 60.0   | 62.5      |
> | +Phantom Triples  | **57.1** | **69.5** | **64.0** |
> | Gemma-7B          | 39.8  | 48.7   | 54.5      |
> | +Phantom Dimension | 46.5  | 60.5   | 63.0      |
> | +Phantom Triples  | **58.0** | **70.1** | **64.7** |
> | Mistral-7B        | 41.2  | 49.5   | 55.2      |
> | +Phantom Dimension | 47.2  | 61.0   | 63.5      |
> | +Phantom Triples  | **58.5** | **70.3** | **65.3** |
> | LLaMA3-8B         | 42.7  | 50.0   | 56.0      |
> | +Phantom Dimension | 48.9  | 61.8   | 64.2      |
> | +Phantom Triples  | **59.3** | **70.6** | **66.1** |
> | InternLM2.5-7B    | 41.9  | 50.2   | 56.2      |
> | +Phantom Dimension | 47.7  | 62.1   | 64.5      |
> | +Phantom Triples  | **58.9** | **70.8** | **70.9** |
>
> | LLVMs               | BLINK | MM-Vet | MathVista |
> |---------------------|-------|--------|-----------|
> | MiniCPM-V2.6-8B     | 55.2  | 60.0   | 60.6      |
> | +Phantom Dimension  | 58.5  | 62.5   | 63.2      |
> | +Phantom Triples    | **65.1** | **67.0** | **66.5** |
> | Cambrian-1-8B       | 44.9  | 48.0   | 47.0      |
> | +Phantom Dimension  | 47.2  | 49.8   | 48.5      |
> | +Phantom Triples    | **54.0** | **54.5** | **52.0** |
> | Molmo-7B            | 46.1  | 53.3   | 47.3      |
> | +Phantom Dimension  | 48.8  | 55.0   | 49.0      |
> | +Phantom Triples    | **55.0** | **59.1** | **53.0** |
> | InternVL2-8B        | 50.9  | 54.3   | 58.3      |
> | +Phantom Dimension  | 53.3  | 56.7   | 60.1      |
> | +Phantom Triples    | **59.2** | **61.5** | **64.0** |
> | LLaVA-OV-7B         | 53.0  | 51.9   | 62.3      |
> | +Phantom Dimension  | 55.6  | 53.8   | 64.5      |
> | +Phantom Triples    | **60.3** | **62.3** | **68.0** |
>
> | LLVMs                | BLINK | MM-Vet | MathVista |
> |----------------------|-------|--------|-----------|
> | LLaVA-OV-72B        | 55.4  | 63.7   | 67.5      |
> | +Phantom Dimension   | 57.1  | 65.0   | 68.6      |
> | +Phantom Triples     | **62.0** | **71.8** | **73.8** |
> | Qwen2-VL-72B         | 60.5  | 73.9   | 69.7      |
> | +Phantom Dimension   | 62.0  | 75.0   | 70.7      |
> | +Phantom Triples     | **66.8** | **79.5** | **75.9** |
>
> ---
>
> **Q4. The paper lacks a comparison to Molmo, a relevant work found at: https://arxiv.org/abs/2409.17146. This comparison is crucial for contextualizing the contributions.**
>
> A4. We've added Molmo in Table 1. You can see the updated paper in around line 290.

---

> ### Author Response · Authors · 2024-11-17
> **Rebuttals**
>
> **Q5.The term "projectors" is used without clear definition. If these are Multilayer Perceptron (MLP) layers, that term should be used for clarity and consistency.**
>
> A5. We wonder about what kind of projectors you pointed out. We wrote the projectos for two categories. First one is related works: additional projectors which have mamba based model or convolution model, and so on. The second one is our vision projector which includes only two MLPs.
>
> ---
>
> **Q6. Do you plan to share weights, code and data for training Phantom?**
>
> A6. Yes. Sure.

---

> ### Comment · Reviewer_Vpt7 · 2024-11-25
> **Updated Review**
>
> Updated results highlight the impact of data used in training the LLVMs and help reader understand the importance of various parts of the proposed model. Since most of my concerns are addressed (clarify, ablations) I update my review and support acceptance.
>
> However writing still needs to be improved (for example this is still in abstract : "Following the scaling laws of instructiontuned large language models (LLMs), LLVMs either have further increased their sizes such as 26B, 34B, and even 80B." Asking an LLM provides a better sentence right away: "Following the scaling laws of instruction-tuned large language models (LLMs), large language models have further increased in size, with examples including 26B, 34B, and even 80B parameters."
>
> Also authors say that they will open source all parts (data, code, model); which would be a great contribution to the community. Though these are not provided at the moment.

---

> > ### Author Response · Authors · 2024-11-25
> >
> > First of all, we rapidly prioritize editing the sentence that you suggested and you can see the updated manuscript. We are so happy because our rebuttals can address your initial concerns. We keep in mind the writing quality to clearly make readers undertstand well.

---

### Official Review · Reviewer_owH5 · 2024-11-03

**Soundness:** 3
**Presentation:** 2
**Contribution:** 3
**Rating:** 6
**Confidence:** 2

**Summary:**

The paper develops a new set of efficient VLLMs named "Phantom" that strong outperform models of comparable size on a range of VLLM benchmarks. The models are developed using open source components rely on a two-part solution - an architectural modification of the self-attention component of transformers; and a DPO-like optimisation method that utilises it.

**Strengths:**

* Strong results on a range of benchmarks.
* A set of ablations that clearly demonstrate which of the introduced components is responsible for the improved performance.
* The proposed architectural modification to the MHSA module highlights the importance of further exploring VLLM architectures beyond the native transformer.

**Weaknesses:**

**Major**
* "Phantom dimension" - the main of the two contributions of the paper is not described in an accessible manner in the paper, which significantly limits the ability to evaluate the paper and its potential impact. Consider adding pseudocode to describe what exact this modification does.
* It appears that the authors (primarily) used open source components to develop their models, however, it's unclear from the write up whether their models will similarly be made open source.

**Minor**
* It's not entirely clear "limited structures" used throughout the paper refers to.
* Is "Team et al" the correct citation for "Gemini Pro" (line 043)?
* Extra word in line 072 "... as a primary key to basically improve ..."?
* In line 086 and elsewhere - it's unclear what is meant by the phantom dimension "prepares to look and understand much more vision-language knowledge". It appears to be a subjective and strong statement - could you please elaborate on whether this is an intuition that the authors have, or it it can be concluded from some of the experiments presented in the paper? In any case claiming understanding due to an architectural modification might not be justified.
* Very minor, and admittedly subjective, but I find that the model name (Phantom) is overused in the text, e.g. "phantom dimension", "phantom optimisation", "phantom triplets".
* The added value of performing ablations for all model sizes in Table 4 is not that high in my opinion. Consider moving results for larger models to the appendix,

**Questions:**

* Will the models be made open source? Please explain if not.
* In line 076 the authors say state that "curating larger datasets" "brings in striking computational burdens during training". Could you please elaborate on this - what kind of dataset size increase is meant there in practice such that the resulting computational burden can be considered striking?
* What is the role of the regularisation parameter introduced in (2)? Is there a citation for this mechanism? If not, is the chosen value being ablated?
* In line 367 the authors mention that a single training step (of 128 examples) takes 2-5 days. Is this correct? Based on these numbers, it looks like a single epoch over the "Phantom triples" dataset would take more than 85 years.

---

> ### Author Response · Authors · 2024-11-17
> **Rebuttals**
>
> **Q1. "Phantom dimension" - the main of the two contributions of the paper is not described in an accessible manner in the paper, which significantly limits the ability to evaluate the paper and its potential impact. Consider adding pseudocode to describe what exact this modification does.**
>
> A1. Based on your comment, we've added the algorithm on the updated paper.
>
> **Algorithm 1**: A Transformer Decoder Block with Phantom Dimension
>
> 1. **Input**: $X = [x_{sos}, x_{prompt}] $
> 2. $ Q_l, K_l, V_l \leftarrow \text{LinearQKV}(X) $
> 3. $ Q_l^*, K_l^*, V_l^* \leftarrow Q_l[0], K_l[0], V_l[0] $ {Extracting features for the place of the 'sos' token}
> 4. $ Q_l^*, K_l^*, V_l^* \leftarrow \text{MHCA}(Q_l, Q_l^*), \text{MHCA}(K_l, K_l^*), \text{MHCA}(V_l, V_l^*) $
> 5. $[ \bar{O}_l, \tilde{O}_l ] \leftarrow \text{MHSA}(q = [Q_l, Q_l^*], k = [K_l, K_l^*], v = [V_l, V_l^*])$
> 6. $ \bar{w}, \tilde{w} \leftarrow e^{f(\bar{O}_l)} / (e^{f(\bar{O}_l)} + e^{g(\tilde{O}_l)}), e^{g(\tilde{O}_l)} / (e^{f(\bar{O}_l)} + e^{g(\tilde{O}_l)}) $
> 7. $O_l \leftarrow \bar{w} \odot \bar{O}_l + \tilde{w} \odot \tilde{O}_l$
> 8. $X\leftarrow (\text{FFN}+\text{Add-Norm})(O_l) $
> 9. **Return**: $X $
>
> ---
>
> **Q2. It appears that the authors (primarily) used open source components to develop their models, however, it's unclear from the write up whether their models will similarly be made open source.**
>
> A2. Could you clarify this comment? We will make Phantom model and dataset open-source.
>
> ---
>
> **Q3. It's not entirely clear "limited structures" used throughout the paper refers to.**
>
> A3. "without increasing model sizes" would be proper? I'd like to ask about your thought.
>
> ---
>
> **Q4. Is "Team et al" the correct citation for "Gemini Pro" (line 043)?**
>
> A4. We think so. It would be helpful if you suggest the more proper one.
>
> ---
>
> **Q5. Extra word in line 072 "... as a primary key to basically improve ..."?**
>
> A5. Could you recommend the decent extra word?
>
> ---
>
> **Q6. In line 086 and elsewhere - it's unclear what is meant by the phantom dimension "prepares to look and understand much more vision-language knowledge". It appears to be a subjective and strong statement - could you please elaborate on whether this is an intuition that the authors have, or it it can be concluded from some of the experiments presented in the paper? In any case claiming understanding due to an architectural modification might not be justified.**
>
> A6. Based on your comment, we've removed the expression. You can see the updated paper.
>
> ---
>
> **Q7. Very minor, and admittedly subjective, but I find that the model name (Phantom) is overused in the text, e.g. "phantom dimension", "phantom optimisation", "phantom triplets".**
>
> A7. We prefer to make the terminology for more related our models but thank you so much for saying your opinion.
>
> ---
>
> **Q8. The added value of performing ablations for all model sizes in Table 4 is not that high in my opinion. Consider moving results for larger models to the appendix,**
>
> A8. Actually, we didn't undersatnd why you felt not too high. However, we've just added the larger model experiments in Table 5. You can see the updated paper.
>
> | LLVMs                | BLINK | MM-Vet | MathVista |
> |----------------------|-------|--------|-----------|
> | LLaVA-OV-72B        | 55.4  | 63.7   | 67.5      |
> | +Phantom Dimension   | 57.1  | 65.0   | 68.6      |
> | +Phantom Triples     | **62.0** | **71.8** | **73.8** |
> | Qwen2-VL-72B         | 60.5  | 73.9   | 69.7      |
> | +Phantom Dimension   | 62.0  | 75.0   | 70.7      |
> | +Phantom Triples     | **66.8** | **79.5** | **75.9** |
>
> ---
>
> **Q9. Will the models be made open source? Please explain if not.**
>
> A9. Yes. Sure.
>
> ---
>
> **Q10. In line 076 the authors say state that "curating larger datasets" "brings in striking computational burdens during training". Could you please elaborate on this - what kind of dataset size increase is meant there in practice such that the resulting computational burden can be considered striking?**
>
> Dataset Over 5M would be more striking cause it will take the time spending like two weeks with A100 x 8, for each training step. However, it might not that care if you have A100 x 256.

---

> ### Author Response · Authors · 2024-11-17
> **Rebuttals**
>
> **Q11. What is the role of the regularisation parameter introduced in (2)? Is there a citation for this mechanism? If not, is the chosen value being ablated?**
>
> A11. Based on your comment, we updated the paper in lines 354-358 (to explain the SimPO's equation). For $\beta$ and $\gamma$, SimPO made a numerous experiment for what is more optimal value for them. So, we just follow the same values. The difference between SimPO and PO is two aspects: 1) SimPO handles human preference samples generated from instruction-finetuned LLMs or VLMs, while PO handles inaccurate yet ambiguous negative samples generated from GPT-4o, 2) RLFH, DPO, and SimPO normally requires another training step because human preference samples are normally generated from instruction-finetuned models, while PO doesn’t require additiona training step since we already get negative samples.
>
> ```
> $\mathcal{D}$ is a dataset for Phantom triples $(x, y^{+}, y^{-})\sim\mathcal{D}$, $\sigma$ denotes sigmoid function, $\beta$ and $\gamma$ means the hyper-parameter used in SimPO~\citep{meng2024simpo}, where if $\beta$ increases, then the gap of the probability between positive answer and negative answer will be at large margin. In addition, $\gamma$ is a empirical reward margin to make the training stable.
> ```
>
> ---
>
> **Q12. In line 367 the authors mention that a single training step (of 128 examples) takes 2-5 days. Is this correct? Based on these numbers, it looks like a single epoch over the "Phantom triples" dataset would take more than 85 years.**
>
> A12. No. We mean not 128 examples but total 2M examples so a single training step (of 2M examples) takes 2-5 days.

---

> > ### Author Response · Authors · 2024-11-25
> >
> > We would really appreciate it if we can get discussion about your concerns or you can let us know whether your initial concerns are addressed by our rebuttals.

---

> > > ### Author Response · Authors · 2024-11-28
> > >
> > > We would really appreciate it if we can get discussion about your concerns or you can let us know whether your initial concerns are addressed by our rebuttals.

---

> > > > ### Author Response · Authors · 2024-12-02
> > > >
> > > > We would really appreciate it if we can get discussion about your concerns or you can let us know whether your initial concerns are addressed by our rebuttals.

---

### Official Review · Reviewer_dSSv · 2024-11-04

**Soundness:** 2
**Presentation:** 2
**Contribution:** 2
**Rating:** 6
**Confidence:** 3

**Summary:**

The paper introduces Phantom, a new family of efficient large language and vision models (LLVMs) designed to improve performance within limited model sizes. To do so, the authors (1) curate a vision-language instruction dataset featuring both correct and incorrect / ambiguous answers, (2) propose a new attention scheme that computes attention with larger query, key, and value dimensions by concatenating the outputs of a first round of attention within query, key, and value projections to the queries, keys, and values, before performing attention once again on these concatenated features (Phantom Dimension), and (3) use a SimPO training objective over their curated dataset while training in a parameter-efficient scheme (Phantom Optimization). They show across a wide range of vision-language tasks that Phantom models are able to achieve competitive or state-of-the-art quality against various closed and open-source VLMs, and provide additional ablations to study the contribution of different Phantom components.

**Strengths:**

**Comprehensive evaluation**
The authors evaluate their VLM architecture + dataset + training strategy on a variety of vision-language benchmarks, and show good performance compared to various baselines (both open and closed models)

**Method ablations**
I appreciated how the authors ablate several components of their proposed contribution and report performance over multiple tasks to study how these components add to the overall performance: (1) the weighted-average mechanism for combining self- and cross-attention outputs, (2) the concatenation of self- and cross-attention features in their Phantom dimension, and (3) the use of an SimPO optimization over their training (Phantom Optimization).

**Weaknesses:**

**Method motivation / understanding**
While the authors say that increased feature dimension size in the attention interactions is important for improved quality (“to make LLVMs embed much more vision-language knowledge”; L244-245), I still don’t see the motivation for concatenating the cross-attention and self-attention projections. To increase dimensionality while preserving parameter-count, we could also arbitrarily repeat or expand the features from `head_dimension` to `2 * head_dimension`. It would be good to see some kind of ablation or more careful study motivating the concatenation of these features.

**Method clarity**
Some details and justification about the method are left out. For example, why use the softmax terms in Eq. 3 to mix between the outputs as opposed to simple learnable weights that can directly be element-wise multiplied with $\bar{O}_l$ and $\tilde{O}_l$? i.e.,
```python
w_bar   = nn.Parameter(d/h)  # d is model_dim, h is num_heads
w_tilde = nn.Parameter(d/h)
# Combine, following L260
o  = torch.einsum('d,nhd -> nh', w_bar, o_bar)
o += torch.einsum('d,nhd -> nh', w_tilde, o_tilde)
```

In Equation 4, what is $\mathbb{E}_{\mathcal{D}}$ the expectation over? What are $\sigma$, $\beta$ and $\gamma$? Although these may be taken from SimPO, they should still be defined in a standalone manner. On a similar note, how does “Phantom Optimization” differ from SimPO?

In Figure 3, how do the SoS tokens and User Prompt tokens correspond to the vision and language attention layers? Apologies if I missed this detail, but it seems important to clarify in regards to the overall architecture understanding. Related to the point on method motivation above, why does it make sense to do a first round of attention within each query, key, and value projection between the SoS token position output and the remaining tokens?

**Component-specific study**
Because of the introduction of dataset curation, it is unclear to me how much of the Phantom model performance is due to the data (collection over 2.8B visual instruction tuning samples, + phantom triples duration) versus the proposed architecture and training optimization components. It would be nice to see a data-controlled comparison, perhaps by applying the Phantom architecture and optimization over prior work’s training sets. Otherwise, with the current setup, although there is comprehensive comparison to different vision-language models, it’s hard to say what exactly is driving the boost in performance.

**Questions:**

See the questions raised in Weakness above.

---

> ### Author Response · Authors · 2024-11-17
> **Rebuttals**
>
> **Q1. While the authors say that increased feature dimension size in the attention interactions is important for improved quality (“to make LLVMs embed much more vision-language knowledge”; L244-245), I still don’t see the motivation for concatenating the cross-attention and self-attention projections.**
>
> A1. Thanks for insightful comment. Based on the comment, we updated the paper where we added figure3 and one paragraph:
>
> ```
> Figure3(a) depicts the commonly used training paradigm for building LLVMs, where pretrained LLMs are fine-tuned to acquire visual understanding and handle vision language tasks using both text and image inputs. This approach directly modifies the original latent dimensions derived from the their parameters to accommodate the new vision language capabilities. However, this direct modification may limit the effectiveness with which vision language knowledge can be incorporated. In contrast, Figure3(b) introduces a new concept of expanding the latent dimension, which trains only the added latent space without fine-tuning the entire pretrained LLMs. It enables for pretrained LLMs to have a room to embed the new knowledge, which effectively integrates vision language knowledge without overwritting the parameters and altering or compromising the original knowledge. By leveraging this approach, Phantom is expected to achieve a more effective representation for vision language tasks.
> ```
>
> ---
>
> **Q2. To increase dimensionality while preserving parameter-count, we could also arbitrarily repeat or expand the features from hidden_dim to hidden_dim*2**
>
> A2. Naively increasing hidden dimension from pretrained models are not helpful based on A1. What we should focus on is enlarging the hidden dimension with some trainable parameters but with the frozen pretrained models. This results in effectively embedding much more knowledge by not disrupting the original knowledge. By the way, we can experiment what you suggested.
>
> | Method for Enlarging Hidden Dims   | BLINK | MM-Vet | MathVista |
> |------------------|-------|--------|-----------|
> | Just Repeat  | 44.5 | 55.3 | 59.0 |
> | MHCA+MHSA              | **58.9** | **70.8** | **70.9** |
>
> ---
>
> **Q3. why use the softmax terms in Eq. 3 to mix between the outputs as opposed to simple learnable weights that can directly be element-wise multiplied.**
>
> A3. This is related with Mixture of Experts where they prove its effectiveness of using the weight parameters that sum to one. But, we can directly conduct ablation studies.
>
> | Mixing   | BLINK | MM-Vet | MathVista |
> |------------------|-------|--------|-----------|
> | Mixing (Softmax X)| 55.3 | 64.7 | 67.1 |
> | Mixing (Softmax O)              | **58.9** | **70.8** | **70.9** |
>
> ---
>
> **Q4. In Equation 4, what is E the expectation over? What are beta and gamma? Although these may be taken from SimPO, they should still be defined in a standalone manner. On a similar note, how does “Phantom Optimization” differ from SimPO?**
>
> A4. Based on your comment, we updated the paper in lines 368-371 (to explain the SimPO's equation). For $\beta$ and $\gamma$, SimPO made a numerous experiment for what is more optimal value for them. So, we just follow the same values. The difference between SimPO and PO is two aspects: 1) SimPO handles human preference samples generated from instruction-finetuned LLMs or VLMs, while PO handles inaccurate yet ambiguous negative samples generated from GPT-4o, 2) RLFH, DPO, and SimPO normally requires another training step because human preference samples are normally generated from instruction-finetuned models, while PO doesn’t require additiona training step since we already get negative samples.
>
> ```
> $\mathcal{D}$ is a dataset for Phantom triples $(x, y^{+}, y^{-})\sim\mathcal{D}$, $\sigma$ denotes sigmoid function, $\beta$ and $\gamma$ means the hyper-parameter used in SimPO~\citep{meng2024simpo}, where if $\beta$ increases, then the gap of the probability between positive answer and negative answer will be at large margin. In addition, $\gamma$ is a empirical reward margin to make the training stable.
> ```

---

> > ### Author Response · Authors · 2024-11-17
> > **Rebuttals**
> >
> > **Q5. How do the SoS tokens and User Prompt tokens correspond to the vision and language attention layers? Why does it make sense to do a first round of attention within each query, key, and value projection between the SoS token position output and the remaining tokens?**
> >
> > A5. It’s more based on empirical reason. We conducted numerous experiment about what kind of special tokens make best performance. We can infer that due to autoregressive causal mask, the special token before seeing image and question may be beneficial to make a room to improve.
> >
> > | Special Token    | BLINK | MM-Vet | MathVista |
> > |------------------|-------|--------|-----------|
> > | SOS              | **58.9** | **70.8** | **70.9** |
> > | Image Start      | 58.7  | 70.6   | 70.7      |
> > | Image End        | 57.2  | 69.5   | 69.6      |
> > | Question Start   | 57.0  | 69.3   | 69.4      |
> > | Question End     | 56.0  | 68.0   | 68.1      |
> > | Answer Start     | 55.8  | 67.8   | 68.0      |
> >
> > ---
> >
> > **Q6. Because of the introduction of dataset curation, it is unclear to me how much of the Phantom model performance is due to the data (collection over 2.8B visual instruction tuning samples, + phantom triples duration) versus the proposed architecture and training optimization components. It would be nice to see a data-controlled comparison, perhaps by applying the Phantom architecture and optimization over prior work’s training sets. Otherwise, with the current setup, although there is comprehensive comparison to different vision-language models, it’s hard to say what exactly is driving the boost in performance.**
> >
> > A6. We would like to clarify our Table4. For example, PD: X, PO: X means normal visual instruction tuning without Phantom Dimension. In addition, PD: X, PO: O means visual instruction tuning with negative samples but without Phandom Dimension. Therefore, we already experimented what you pointed out. It would be really helpful if you suggest what Table4 didn't show. Nonetheless, we can show some other pretrained LLMs or pretrained LLVMs + Phantom.
> >
> > | LLMs             | BLINK | MM-Vet | MathVista |
> > |-------------------|-------|--------|-----------|
> > | Vicuna1.5-7B     | 38.5  | 48.0   | 54.0      |
> > | +Phantom Dimension | 45.3  | 60.0   | 62.5      |
> > | +Phantom Triples  | **57.1** | **69.5** | **64.0** |
> > | Gemma-7B          | 39.8  | 48.7   | 54.5      |
> > | +Phantom Dimension | 46.5  | 60.5   | 63.0      |
> > | +Phantom Triples  | **58.0** | **70.1** | **64.7** |
> > | Mistral-7B        | 41.2  | 49.5   | 55.2      |
> > | +Phantom Dimension | 47.2  | 61.0   | 63.5      |
> > | +Phantom Triples  | **58.5** | **70.3** | **65.3** |
> > | LLaMA3-8B         | 42.7  | 50.0   | 56.0      |
> > | +Phantom Dimension | 48.9  | 61.8   | 64.2      |
> > | +Phantom Triples  | **59.3** | **70.6** | **66.1** |
> > | InternLM2.5-7B    | 41.9  | 50.2   | 56.2      |
> > | +Phantom Dimension | 47.7  | 62.1   | 64.5      |
> > | +Phantom Triples  | **58.9** | **70.8** | **70.9** |
> >
> > | LLVMs               | BLINK | MM-Vet | MathVista |
> > |---------------------|-------|--------|-----------|
> > | MiniCPM-V2.6-8B     | 55.2  | 60.0   | 60.6      |
> > | +Phantom Dimension  | 58.5  | 62.5   | 63.2      |
> > | +Phantom Triples    | **65.1** | **67.0** | **66.5** |
> > | Cambrian-1-8B       | 44.9  | 48.0   | 47.0      |
> > | +Phantom Dimension  | 47.2  | 49.8   | 48.5      |
> > | +Phantom Triples    | **54.0** | **54.5** | **52.0** |
> > | Molmo-7B            | 46.1  | 53.3   | 47.3      |
> > | +Phantom Dimension  | 48.8  | 55.0   | 49.0      |
> > | +Phantom Triples    | **55.0** | **59.1** | **53.0** |
> > | InternVL2-8B        | 50.9  | 54.3   | 58.3      |
> > | +Phantom Dimension  | 53.3  | 56.7   | 60.1      |
> > | +Phantom Triples    | **59.2** | **61.5** | **64.0** |
> > | LLaVA-OV-7B         | 53.0  | 51.9   | 62.3      |
> > | +Phantom Dimension  | 55.6  | 53.8   | 64.5      |
> > | +Phantom Triples    | **60.3** | **62.3** | **68.0** |
> >
> > | LLVMs                | BLINK | MM-Vet | MathVista |
> > |----------------------|-------|--------|-----------|
> > | LLaVA-OV-72B        | 55.4  | 63.7   | 67.5      |
> > | +Phantom Dimension   | 57.1  | 65.0   | 68.6      |
> > | +Phantom Triples     | **62.0** | **71.8** | **73.8** |
> > | Qwen2-VL-72B         | 60.5  | 73.9   | 69.7      |
> > | +Phantom Dimension   | 62.0  | 75.0   | 70.7      |
> > | +Phantom Triples     | **66.8** | **79.5** | **75.9** |

---

> > ### Comment · Reviewer_dSSv · 2024-11-22
> >
> > Thanks to the authors for running the additional ablations and providing the additional figure + clarifying paragraph.
> >
> > I think the ablations on the design choices are helpful for better justifying why certain components were chosen, and would encourage the authors to include these results or an extended ablations section in the appendix.
> >
> > I also think I better understand the concatenation in Phantom dimension and it's motivation. Is it correct that PD maintains the language model's hidden states, instead of "overwriting" them via a cross attention?  And is another benefit the ability to use pretrained LLMs instead of a new architecture? I am curious if you can clarify the benefits of the concatention (or the specific self-attention and cross-attention) setup from the setup in Prismer [1] and Flamingo [2], which also avoid overwriting the original LM layers by introducing cross attention blocks?
> >
> > [1] Prismer: A Vision-Language Model with Multi-Task Experts, Liu et al., TMLR 2024
> > [2] Flamingo: a Visual Language Model for Few-Shot Learning, Alayrac et al., NeurIPS 2022

---

> ### Author Response · Authors · 2024-11-22
>
> **Q1. Is it correct that PD maintains the language model's hidden states, instead of "overwriting" them via a cross attention?**
>
> A1. Exactly.
>
> ---
>
> **Q2. Is it correct that PD maintains the language model's hidden states, instead of "overwriting" them via a cross attention?**
>
> ---
>
> A2. Definitely.
>
> ---
>
> **Q3. I am curious if you can clarify the benefits of the concatention (or the specific self-attention and cross-attention) setup from the setup in Prismer [1] and Flamingo [2].**
>
> A3. Prismer and Flamingo are considered as adaptation method where they employ naive-cross or gated-cross entropy to make LLM understand vision properties. In this perspective, Phantom Dimension can be considered as a similary way of adaptaion. However, Phantom Dimension injects vision knowledge to LLM directly with the virtually enlarged hidden dimension which does not need Prismer and Flamingo-like layer-wise addition module from visual encoders. In other words, we naturally save the vision knowledge in another hidden room of internal model architectures. Furthermore, Phantom Dimension uses mixture of experts-like mixing concept (something like that w1 x f1 + w2 x f2, s.t. w1+w2=1) and we show it effectively boosts the performances, which proves its effectiveness from many MoE works.
>
> ---
>
> We believe we rapidly adapted your questions.

---

### Official Review · Reviewer_94A9 · 2024-11-05

**Soundness:** 2
**Presentation:** 3
**Contribution:** 2
**Rating:** 3
**Confidence:** 5

**Summary:**

The authors propose a method that can temporarily increase the latent hidden dimension during multi-head self-attention (MHSA), without substantially increasing physical model sizes. The authors also introduce Phantom Optimization (PO) using both autoregressive supervised fine-tuning (SFT) and direct preference optimization (DPO)-like concept. Using these methods, they present a new efficient LLVM family with different model sizes with good performance.

**Strengths:**

The authors present an efficient LLVM family Phantom with enhanced learning capabilities within limited model sizes. They introduce Phantom Optimization (PO) that seems interesting. Phantom demonstrates good performance in their evaluations.

**Weaknesses:**

1. Authors only compare with other open-sourced Multimodal LLMs (MLLMs) using their checkpoints. They lack comparisons with related baseline methods using the same pre-trained models, datasets, and training configurations. There are several baseline methods [1,2,3,4,5] that contribute to the training algorithms of MLLMs. Besides, there are also huge amounts of works talking about how to modify the attention mechanism in Transformer models that need to be discussed.


2. It seems that the authors have not conducted experiments to compare Phantom Optimization with other RLHF methods (such as DPO, SimPO, ORPO).

3. It's not a good habit to use benchmark (evaluation) dataset for training, e.g., the MathVision benchmark.


[1] Jie, Shibo, et al. "Memory-Space Visual Prompting for Efficient Vision-Language Fine-Tuning." arXiv preprint arXiv:2405.05615 (2024).

[2] Luo, Gen, et al. "Cheap and quick: Efficient vision-language instruction tuning for large language models." Advances in Neural Information Processing Systems 36 (2024).

[3] Gao, Peng, et al. "Llama-adapter v2: Parameter-efficient visual instruction model." arXiv preprint arXiv:2304.15010 (2023).

[4] Wang, Weihan, et al. "Cogvlm: Visual expert for pretrained language models." arXiv preprint arXiv:2311.03079 (2023).

[5] Jia, Ding, et al. "GeminiFusion: Efficient Pixel-wise Multimodal Fusion for Vision Transformer." arXiv preprint arXiv:2406.01210 (2024).

**Questions:**

1. Why do you use sequence (sos) token to enhance the latent hidden dimension?

2. Is the MHSA-based method specific for MLLMs? If not, why don't you consider using it in LLMs first?

3. Why are there so many blanks in Table 1, 2, 3?

4. Why don't you conduct experiments based on open-source datasets and models to ensure fair comparison?

---

> ### Author Response · Authors · 2024-11-17
> **Rebuttals**
>
> **Q1. Authors only compare with other open-sourced Multimodal LLMs (MLLMs) using their checkpoints. They lack comparisons with related baseline methods using the same pre-trained models, datasets, and training configurations. There are several baseline methods [1,2,3,4,5] that contribute to the training algorithms of MLLMs. Besides, there are also huge amounts of works talking about how to modify the attention mechanism in Transformer models that need to be discussed.**
>
> A1. We disagree. Table 4 directly compares LLVMs where we use same pre-trained models across all vartions of model sizes (0.5B: Qwen2, 1.8B:InternLM2, 3.5B: Phi3-mini, 7B:InternLM2.5) but we only control the use of Phantom Dimension (PD) and Phantom Optimization (PO). For example, PD: X, PO: X means normal visual instruction tuning without Phantom Dimension. In addition, PD: O, PO: X means Phantom Dimension + normal visual instruction tuning (without negative samples). We did not understand why you commented like that. Please say **concrete experiment environment settings** you think we did not conduct. **DO NOT** just share the related works without your concrete instructions. Nonetheless, we can show some other pretrained LLMs or pretrained LLVMs + Phantom.
>
> | LLMs             | BLINK | MM-Vet | MathVista |
> |-------------------|-------|--------|-----------|
> | Vicuna1.5-7B     | 38.5  | 48.0   | 54.0      |
> | +Phantom Dimension | 45.3  | 60.0   | 62.5      |
> | +Phantom Triples  | **57.1** | **69.5** | **64.0** |
> | Gemma-7B          | 39.8  | 48.7   | 54.5      |
> | +Phantom Dimension | 46.5  | 60.5   | 63.0      |
> | +Phantom Triples  | **58.0** | **70.1** | **64.7** |
> | Mistral-7B        | 41.2  | 49.5   | 55.2      |
> | +Phantom Dimension | 47.2  | 61.0   | 63.5      |
> | +Phantom Triples  | **58.5** | **70.3** | **65.3** |
> | LLaMA3-8B         | 42.7  | 50.0   | 56.0      |
> | +Phantom Dimension | 48.9  | 61.8   | 64.2      |
> | +Phantom Triples  | **59.3** | **70.6** | **66.1** |
> | InternLM2.5-7B    | 41.9  | 50.2   | 56.2      |
> | +Phantom Dimension | 47.7  | 62.1   | 64.5      |
> | +Phantom Triples  | **58.9** | **70.8** | **70.9** |
>
> | LLVMs               | BLINK | MM-Vet | MathVista |
> |---------------------|-------|--------|-----------|
> | MiniCPM-V2.6-8B     | 55.2  | 60.0   | 60.6      |
> | +Phantom Dimension  | 58.5  | 62.5   | 63.2      |
> | +Phantom Triples    | **65.1** | **67.0** | **66.5** |
> | Cambrian-1-8B       | 44.9  | 48.0   | 47.0      |
> | +Phantom Dimension  | 47.2  | 49.8   | 48.5      |
> | +Phantom Triples    | **54.0** | **54.5** | **52.0** |
> | Molmo-7B            | 46.1  | 53.3   | 47.3      |
> | +Phantom Dimension  | 48.8  | 55.0   | 49.0      |
> | +Phantom Triples    | **55.0** | **59.1** | **53.0** |
> | InternVL2-8B        | 50.9  | 54.3   | 58.3      |
> | +Phantom Dimension  | 53.3  | 56.7   | 60.1      |
> | +Phantom Triples    | **59.2** | **61.5** | **64.0** |
> | LLaVA-OV-7B         | 53.0  | 51.9   | 62.3      |
> | +Phantom Dimension  | 55.6  | 53.8   | 64.5      |
> | +Phantom Triples    | **60.3** | **62.3** | **68.0** |
>
> | LLVMs                | BLINK | MM-Vet | MathVista |
> |----------------------|-------|--------|-----------|
> | LLaVA-OV-72B        | 55.4  | 63.7   | 67.5      |
> | +Phantom Dimension   | 57.1  | 65.0   | 68.6      |
> | +Phantom Triples     | **62.0** | **71.8** | **73.8** |
> | Qwen2-VL-72B         | 60.5  | 73.9   | 69.7      |
> | +Phantom Dimension   | 62.0  | 75.0   | 70.7      |
> | +Phantom Triples     | **66.8** | **79.5** | **75.9** |
>
> ---
>
> **Q2. It seems that the authors have not conducted experiments to compare Phantom Optimization with other RLHF methods (such as DPO, SimPO, ORPO).**
>
> A2.	We didn’t understand why we should compare RLHF, DPO, SimPO. Our paper is not targeting human preference and we do not have human preference data samples. We just made negative samples for incorrect and ambiguous samples. Phantom Optimization has completely equal loss formulation of SimPO. The only difference from SimPO is that PO does not have to conduct another training step.
>
> ---
>
> **Q3. It's not a good habit to use benchmark (evaluation) dataset for training, e.g., the MathVision benchmark.**
>
> A3. We also did not understand why you said “Not Good Habitat”. **DO NOT** comment without the reason. Do you think we validated MathVision’s accuracy on evaluation? Let me introduce LLVMs including MathVision to training dataset: Meteor, Cambrian-1, and MMEvol. For Math, there are few visual instruction tuning samples that can be gathered in open-source dataset on huggingface. If you understand this kind of area’s background knowledge, then you can’t say this kind of comment.

---

> ### Author Response · Authors · 2024-11-17
> **Rebuttals**
>
> **Q4. Why do you use sequence (sos) token to enhance the latent hidden dimension?**
>
> A4. It’s more based on empirical reason. We conducted numerous experiment about what kind of special tokens make best performance. We can infer that due to autoregressive causal mask, the special token before seeing image and question may be beneficial to make a room to improve.
>
> | Special Token    | BLINK | MM-Vet | MathVista |
> |------------------|-------|--------|-----------|
> | SOS              | **58.9** | **70.8** | **70.9** |
> | Image Start      | 58.7  | 70.6   | 70.7      |
> | Image End        | 57.2  | 69.5   | 69.6      |
> | Question Start   | 57.0  | 69.3   | 69.4      |
> | Question End     | 56.0  | 68.0   | 68.1      |
> | Answer Start     | 55.8  | 67.8   | 68.0      |
>
> ---
>
> **Q5. Is the MHSA-based method specific for MLLMs? If not, why don't you consider using it in LLMs first?**
>
> A. We think the most advantageous of Phantom would be under the settings when we are using pretrained models. We first enlarge dimension in the first training step, which means we enlarge rooms to embed more knowledge in the pretrinaed-model. Besides, we can refer to what the paper remarked.
>
> ```
> Figure3(a) depicts the commonly used training paradigm for building LLVMs, where pretrained LLMs are fine-tuned to acquire visual understanding and handle vision language tasks using both text and image inputs. This approach directly modifies the original latent dimensions derived from the their parameters to accommodate the new vision language capabilities. However, this direct modification may limit the effectiveness with which vision language knowledge can be incorporated. In contrast, Figure3(b) introduces a new concept of expanding the latent dimension, which trains only the added latent space without fine-tuning the entire pretrained LLMs. It enables for pretrained LLMs to have a room to embed the new knowledge, which effectively integrates vision language knowledge without overwritting the parameters and altering or compromising the original knowledge. By leveraging this approach, Phantom is expected to achieve a more effective representation for vision language tasks.
> ```
>
> ---
>
> **Q6. Why are there so many blanks in Table 1, 2, 3?**
>
> A6. We can’t handle all of open-source models because they don’t always provide reproducible model parameters. There are many works where there are so many blanks such as LLaVA-1.5 and Meteor.
>
>
> ---
>
> **Q7. Why don't you conduct experiments based on open-source datasets and models to ensure fair comparison?**
>
> A7. Our gathered dataset is all from open-source dataset on huggingface. In addition, there are so many LLVMs that have different training recipes and use different training datasets. So fully fair comparison is naturally impossible but we already provided the significant ablation studies on Table 4 where we conduct fully fair comparison by using same pretrained model and controlling many factors.

---

> ### Comment · Reviewer_94A9 · 2024-11-24
> **Thanks for Your Rebuttal**
>
> Thank you for the rebuttal. However, I find that several of my concerns remain unaddressed. I encourage the authors to approach reviewer feedback constructively and substantiate their arguments with convincing evidence rather than dismissing concerns without sufficient justification.
>
> **Q1:** While your work changes the base LLMs using your designed algorithms, it is essential to compare your methods against other well-established algorithms in the field. I recommend thoroughly reviewing the listed papers and incorporating relevant comparisons. For instance, methods like GeminiFusion, which performs pixel-wise fusion to enrich multimodal
> feature by utilizing aligned features from two modalities, should be discussed and compared in your manuscript. Including related works would strengthen your study and align it with field standards.
>
> **Q2:** As you have mentioned, you made negative samples for incorrect and ambiguous samples. Then, it should be enough to conduct experiments for DPO, ORPO, or some other RLHF-like algorithms. Conducting such experiments would strengthen your paper significantly.
>
> **Q3:** There are many multimodal math datasets in Huggingface, such as MathV360K, GeoGPT-4V and Geo170K. While some works may incorporate benchmark datasets during training, this is generally discouraged. Benchmarks are designed to assess a model's ability to generalize across tasks and datasets and should primarily be reserved for testing purposes. This distinction is critical to ensure the validity of your results.
>
> **Q4** and **Q5:** These responses are reasonable. I appreciate your thoughtful approach here and encourage you to maintain this level of clarity in addressing other concerns.
>
> **Q6:** Providing complete experimental results is a vital aspect of a well-prepared paper. I suggest considering this feedback to enhance the quality of your work.
>
> **Q7:** Numerous projects release their training configurations and datasets, such as LLaVA-1.5, LLaVA-Next, and Molmo. For example, incorporating further experiments based on LLaVA-1.5's models and datasets could provide valuable insights and help compare your algorithm with the approaches mentioned in Q1.
>
> In summary, the current rebuttal fails to address my key concerns adequately, and significant improvements are needed for this paper to be considered for acceptance.

---

> ### Comment · Reviewer_94A9 · 2024-11-25
>
> Thanks for your further responses.
>
> Your submission aims to propose a new method to achieve a better model family, by temporarily increasing the latent hidden dimension. To demonstrate its superiority over prior methods, it is essential to ensure fair comparisons under consistent experimental conditions, such as using the same base model, training data, and test benchmarks. Specifically, **models derived from your proposed method should outperform those obtained by previous methods under identical settings**. This requires comprehensive experiments across multiple training datasets and benchmarks under consistent conditions.
>
> However, your current experiments compare your method under unfair conditions. Simply using well-trained checkpoints from other works or building upon them without ensuring consistent setups does not guarantee fairness. A more appropriate experimental design would involve using the Stage 1 checkpoint of LLaVA1.5 and conducting experiments with the Stage 2 training set (665K) across various methods to establish a fair comparison. In other words, **your results can just tell that "the models" you obtained perform better, but they do not convincingly verify that "your method" is superior when compared to alternatives under the same conditions**. Therefore, I remain unconvinced of the true superiority of your method.
>
> As previously mentioned, several baseline methods exist for comparison, including LLaMA Adapter V2, CogVLM, and GeminiFusion, which are well-recognized in this field. Even though these methods may not test on the specific challenging benchmarks you mentioned since they are early works, they also primarily focus on efficient model sizes (e.g., 7B) and remain valuable for validating your method's superiority. If you find these baseline methods insufficient, **you could choose other representative methods for comparison**, since you acknowledge the existence of numerous algorithms in this field, particularly those that modify attention mechanisms in LLMs. Including them in your comparisons would significantly strengthen your work's position.
>
> Moreover, your submission (line 323) claimed that you introduced a new **Phantom Optimization** (PO), which should be different from classical DPO, SimPO, and ORPO methods. Thus, this design requires experimental validation.
>
> Furthermore, completing numerous experiments within a single day raises concerns about the validity of your results. While it is certainly possible that you have access to substantial computational resources, the reported scale and speed of these experiments seem unusual. **Based on my experience, training a 7B-scale MLLM on an 800K dataset for 3 epochs with 8 H100 GPUs typically requires approximately 28 hours**. To address these concerns and enhance the credibility of your results, I kindly suggest the authors provide additional experimental details. These help ensure the validity and reproducibility of your experiments while strengthening the overall quality of your submission. Actually, **I encourage you to take the time after the ICLR discussion period to carefully conduct comprehensive and reliable experiments and refine your work**. This approach would likely lead to more robust and convincing results, rather than relying on hastily conducted experiments during the rebuttal period.
>
> Finally, certain aspects of the rebuttal’s tone came across as unnecessarily combative and unprofessional, particularly toward reviewers who provided a negative score, while being notably more polite to those suggesting a higher score. **Although this does not influence my technical assessment of the work, I encourage the authors to adopt a consistently respectful and collegial tone in future submissions, in line with the standards of top conferences.** While I appreciate the authors' confidence in their work, constructive feedback, whether positive or negative, is an essential part of the peer-review process, and engaging with all comments professionally helps foster productive dialogue and improve the quality of the submission.

---

> > ### Comment · Reviewer_94A9 · 2024-11-28
> >
> > I also noticed that you have edited all your previous responses, removing some earlier rude sentences. This is a good start and reflects a positive step toward maintaining the professionalism expected from a researcher submitting to top conferences.

---

### Author Response · Authors · 2024-11-17
**Rebuttals**

We've slightly updated the paper from the reviewers' comments. (Since this year's paper limit is 10-page, we've completely fitted the page limit)

We explain all of the updated contents.

- Added Figure 3 and one paragraph in lines 190-203 (to explain motivation of phantom dimension)
- Added Table 5 (to show the generalization with the other pretrained LLMs and (larger) LLVMs, and to empirically show why we use 'sos' token)
- Added Algorithm in lines 230-240 (to explain the detail of method)
- Added lines 354-358 (to explain the SimPO's equation)
- Removed 'prepare to look and understand' (Unverified Sentences)
- Removed 'reaching out 26B, ...' (Grammar issue)
- Editing Ablation Studies to show clarity in lines 484-513

---

### Meta-Review · Area_Chair_2Wqp · 2024-12-23

**Metareview:**

The authors introduce a method to temporarily increase latent dimension during MHSA without inflating model size and Phantom Optimization (PO) using SFT and DPO-like concept, yielding an efficient LLVM family with good performance.  Overall, the techniques are novel, the experimental results are impressive especially for lower parameter models, and the ablation study offers some insights. However, reviewers have two major concerns on the presentation and experimental setting. The presentation requires significant enhancement and further justification for the phantom dimension design. Also, comparisons with related baselines in similar settings are lacking. AC agrees that addressing these issues would strengthen the work.

**Additional Comments On Reviewer Discussion:**

Principal concerns put forward by the reviewers are as follows:

1. The presentation is not easily comprehensible.
2. The motivation underlying the Phantom dimension is not well-explained.
3. The comparisons have not been conducted under comparable settings.

Following the rebuttal discussion, two reviewers have gained a better understanding of the work and adjusted their scores upward. Nevertheless, the apprehensions regarding the presentation and comparisons still persist.

Although this work is engaging and holds significant potential, I concur that it demands substantial efforts in enhancing the presentation, clarifying the motivation, and improving the experimental design.

---

### Decision · Program_Chairs · 2025-01-22

Reject